# **LLCP**: LEARNING LATENT CAUSAL PROCESSES FOR REASONING-BASED VIDEO QUESTION ANSWER

**Guangyi Chen**[†•*]**, Yuke Li**[‡*]**, Xiao Liu**[•]**, Zijian Li**[•]**, Eman Al Suradi**[•]**, Donglai Wei**[‡]**, Kun Zhang**[†•]
[•]Mohamed bin Zayed University of Artificial Intelligence, Abu Dhabi, UAE
[†]Carnegie Mellon University, Pittsburgh PA, USA
[‡]Boston College , Massachusetts, USA

## ABSTRACT

Current approaches to Video Question Answering (VideoQA) primarily focus on cross-modality matching, which is limited by the requirement for extensive data annotations and the insufficient capacity for causal reasoning (e.g. attributing accidents). To address these challenges, we introduce a causal framework for video reasoning, termed Learning Latent Causal Processes (**LLCP**). At the heart of **LLCP** lies a multivariate generative model designed to analyze the spatial-temporal dynamics of objects within events. Leveraging the inherent modularity of causal mechanisms, we train the model through self-supervised local auto-regression eliminating the need for annotated question-answer pairs. During inference, the model is applied to answer two types of reasoning questions: accident attribution, which infers the cause from observed effects, and counterfactual prediction, which predicts the effects of counterfactual conditions given the factual evidence. In the first scenario, we identify variables that deviate from the established distribution by the learned model, signifying the root cause of accidents. In the second, we replace embeddings of previous variables with counterfactual ones, enabling us to forecast potential developments. Once we have identified these cause/effect variables, natural language answers are derived through a combination of grammatical parsing and a pre-trained vision-language model. We assess the efficacy of **LLCP** on both synthetic and real-world data, demonstrating comparable performance to supervised methods despite our framework using no paired textual annotations. The code is available at https://github.com/CHENGY12/LLCP.

## 1 INTRODUCTION

Video Question Answering (VideoQA) aims to comprehend and analyze video content, subsequently providing responses to queries posed in natural language. Leveraging its capability of visual dynamics understanding and cross-modality alignment, it serves as a pivotal component in the field of interactive AI, garnering significant attention in recent years.

Current methods Chen et al. (2023); Cong et al. (2021); Feng et al. (2021); Xu et al. (2021); Le et al. (2020) for VideoQA often treat it as a cross-modality matching task to learn temporal dynamics with annotations, as shown the part (a) of Figure 1. For example, HCRN Le et al. (2020) enables video understanding by matching the visual and textual latent embeddings which are learned with annotated natural language question-answer pairs. However, these methods utilize the likelihood of cross-modal matching to learn the spatial-temporal relations, which may fail to capture the underlying causality of the temporal dynamics, leading to serious issues for the performance of video reasoning systems.

To address this challenge, this paper introduces a novel framework for video reasoning, named **LLCP** (Learning Latent Causal Processes), which utilizes a causality-based approach to uncover the latent causal relations and enhance the VideoQA task. **LLCP** employs a temporal multivariate generative model (TMGM) to understand the underlying causal patterns of the agents in the video events. As illustrated on the right side (b) of Figure 1, we show an example of using our framework for traffic

---

[*]Equal contribution.

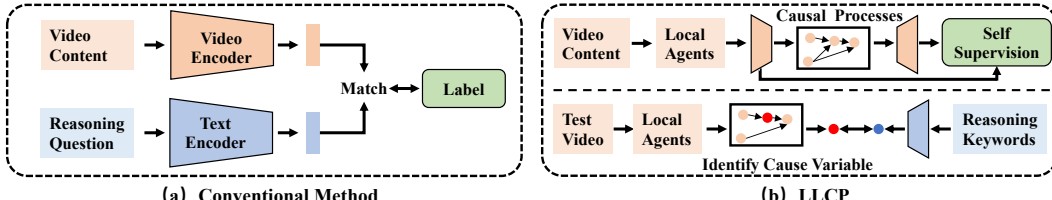

Figure 1: **Comparisons between conventional VideoQA methods and LLCP.** Conventional methods directly match the video content and texts, where the optimization heavily relies on textual annotations. While **LLCP** learns the latent causal processes with a temporal multivariate generative model by self-supervision (without paired data). Taking accident attribution as an example, **LLCP** can answer the reasoning question by matching the identified cause agent with parsed keywords. The red node on the right side denotes the root cause (the agent causing the accident).

accident analysis. We apply the causal mechanisms to model the interactions among agents, and leverage the trained model to conduct causal inference to answer questions.

Specifically, to learn temporal multivariate generative model (TMGM), we first localize the possible objects in the video event. Following the current video reasoning methods Cong et al. (2021); Ji et al. (2020), we utilize a pre-trained tracker or object detector to identify and monitor agents. Subsequently, we employ TMGM to identify the underlying causal mechanisms that guide the behaviors of these agents. Relying on the inherent modularity of causal mechanisms, the TMGM focuses solely on the parent variables from the last time step for each variable. These variables include the agent's historical state, interactions in the vicinity, and environmental information. This generative model is optimized by maximizing the Evidence Lower Bound (ELBO) through self-supervision (without textual annotations).

In the inference, we apply this self-supervised model to answer two types of reasoning questions, including accident attribution, which infers the cause from observed effects, and counterfactual prediction, which predicts the effects of specific counterfactual assumption. For accident attribution, we detect causes by identifying changing factors that do not follow the regular causal processes, i.e., the agents that cannot fit the causal mechanism well, to explain the accident. Specifically, we apply the reconstruction accuracy to check whether an agent is aligned well and then identify the root cause with low reconstruction accuracy. For counterfactual prediction, we first abduct the intra-characters with factual evidence. Then we replace the embeddings of the cause variables with the ones in the counterfactual condition, and predict the potential changes with the learned model. After identifying the cause/effect variables, we apply them to answer the neural language questions. We first implement a grammar parse which uses logical rules to identify the key objects from the question-answer (QA) pairs. Then, leveraging a pre-trained vision-language model, we match the recognized visual objects with the extracted textual concept, thereby answering basic reasoning questions effectively. We evaluated our framework on two simulation datasets and two real-world VideoQA datasets. It consists of SUTD-TrafficQA Xu et al. (2021) for accident attribution and Causal-VidQA Li et al. (2022a) for counterfactual prediction. We show that **LLCP** using no annotated textual question-answer pairs exhibits comparable performance with those methods that train with full supervision.

**Contributions**: 1) We propose a framework, **LLCP**, which introduces a causal perspective of video reasoning to reduce annotation reliance and enhance reasoning ability. 2) We propose TMGM to learn the local latent causal process and apply it for both cause-based and effect-based reasoning tasks. 3) We show that **LLCP** achieves comparable performance with supervised methods without using the paired Question-Answer(QA) annotations for the model training.

## 2 METHOD

In this section, we first introduce the TMGM model and then describe how we can leverage this model in the inference for both cause-based and effect-based video reasoning tasks.

### 2.1 TEMPORAL MULTIVARIATE GENERATIVE MODEL

**Causal Processes.** Without loss of generality, we assume the given video $\mathbf{X}$ consists of $T$ frames and $N$ agents as $\mathbf{X} = \{\mathbf{x}_{t,i}|_{t=1,i=1}^{T,N}\}$. For each variable, we introduce our data-generating pro-

cess shown in Figure 2. In this figure, we provide a closer look using $\mathbf{z}_{t+1,3}$ as the example to explain the causal processes. The gray shade of nodes indicates that the variable is observable. Here we assume that the observed objects $\mathbf{x}$ have the latent variables $\mathbf{z}$. Then the spatial-temporal transitions of the latent variables are divided into three parts: historical states ($\mathbf{z}_{t,3}$), neighborhood interactions ($\mathbf{z}_{t,1}$ and $\mathbf{z}_{t,2}$), and environmental information ($\mathbf{e}_t$). Specifically, For a given variable $\mathbf{x}_{t,i}$, its historical state is the variable with the same identity but in the past frame, i.e., $\mathbf{x}_{t',i}|t' < t$. Similarly, its neighborhoods denote other variables (not experimental variables) in the previous frame, such as $\mathbf{x}_{t',j}|j \neq i, t' < t$. The environment variables denote the environmental context, such as traffic lights, zebra crossing, and traffic signals. Mathematically,

$$\begin{cases} \mathbf{z}_{t,i} & = \mathbf{f}\left(\{\mathbf{z}_{t-\tau,j}|\mathbf{z}_{t-\tau,j} \in \mathbf{Pa}(\mathbf{z}_{t,i})\}, \epsilon_{t,i}\right), \\ \mathbf{x}_{t,i} & = \mathbf{g}(\mathbf{z}_{t,i}), \\ \epsilon_{t,i} & \sim p_{\epsilon_{t,i}}, \end{cases} \quad (1)$$

where the observed object $\mathbf{x}_{t,i}$ is generated from a latent variable $\mathbf{z}_{t,i}$ with a nonlinear function $\mathbf{g}$. We apply a pre-trained multiple object tracking model Pang et al. (2021); Contributors (2020) to detect the objects, which indicates a part of $\mathbf{g}$ and the tracking labels serve as a bridge to connect objects and words. The latent variables $\mathbf{z}_{t,i}$ have stationary time-delayed ($\tau$ denotes the time lag) causal relations which are formulated by $\mathbf{f}$, where $\mathbf{Pa}(\mathbf{z}_{t,i})$ are the parent variables of $\mathbf{z}_{t,i}$, i.e., $\mathbf{Pa}(\mathbf{z}_{t,i})$ denotes the set of latent factors that directly cause $\mathbf{z}_{t,i}$. In addition to parent variables, $\mathbf{z}_{t,i}$ is also influenced by a noise term $\epsilon_{t,i}$, for different spatial-temporal variables, the noise terms $\epsilon_{t,i}$ are mutually-independent. To simplify, and W.L.O.G, in the following sections, we utilize the traffic events as an example to show the model and consider the setting with $\tau = 1$.

Figure 2: **The multivariate causal generation process, in which the historical state $\mathbf{z}_{t,3}$, neighborhoods $\mathbf{z}_{t,1}, \mathbf{z}_{t,2}$, and environment variables $\mathbf{e}_t$ are variables in the temporal dynamic system**

In this scenario, we further finalize the formulation of parent variables $\mathbf{Pa}(\mathbf{z}_{t,i})$ as $\mathbf{Pa}(\mathbf{z}_{t,i}) = \{\mathbf{z}_{t-1,i}, \mathcal{N}(\mathbf{z}_{t-1,i}), \mathbf{e}_{t-\tau}\}$, where $\mathbf{z}_{t-\tau,i}$ denotes the historical state of the current agent. $\mathcal{N}(\mathbf{z}_{t-1,i})$ denotes the neighborhood variables in the $t-1$ time step, which can be easily obtained by the distance on spatial locations. $\mathbf{e}_{t-1}$ is the historical environmental state.

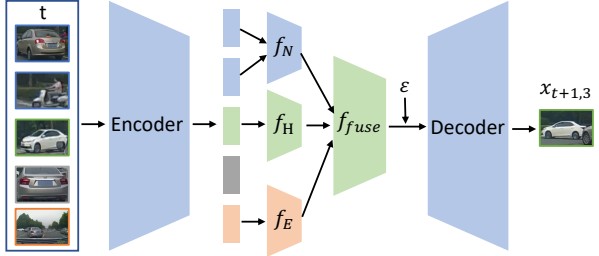

Based on the aforementioned data generation process, we further provide the theoretical analysis to show that the Granger Causal relationship can be captured. We begin with the definition of Granger non-causality of multi-type object sequences.

**Definition 1** *(Granger non-causality of multi-type object sequences) Suppose that the object sequences $\mathbf{X}$ are generated according to Equation 1, we can*

Figure 3: **Network architecture of the temporal multivariate generative model.** We provide an example to show how the temporal multivariate generative model learns the latent causal processes of $\mathbf{x}_{t+1,3}$. Blue denotes the neighborhood objects, green denotes the historical state, gray denotes no-related variables, and orange denotes the environment variable. $f_H$, $f_N$, and $f_E$ are the sub-networks to learn the causal processes from historical states, neighborhood interactions, and environmental clues, respectively. $f_{fuse}$ is a sub-network to fuse them.

*determine the Granger non-causality of the object $\mathbf{x}_i$ with respect to object $\mathbf{x}_j$ as follows: For all $\mathbf{x}_{t-1:t-\tau,1}, \mathbf{x}_{t-1:t-\tau,2}, \cdots, \mathbf{x}_{t-1:t-\tau,N}$, and the same variable with different values $\mathbf{x}_{t-1:t-\tau,i} \neq \mathbf{x}'_{t-1:t-\tau,i}$, if the following condition holds:*

$$\phi_j(\mathbf{x}_{t-1:t-\tau,1}, \cdots, \mathbf{x}_{t-1:t-\tau,i}, \cdots, \mathbf{x}_{t-1:t-\tau,N}) = \phi_j(\mathbf{x}_{t-1:t-\tau,1}, \cdots, \mathbf{x}'_{t-1:t-\tau,i}, \cdots, \mathbf{x}_{t-1:t-\tau,N}),$$

$$(2)$$

*that is, $\mathbf{x}_{t,j}$ is invariant to $\mathbf{x}_{t-1:t-\tau,i}$ with $\phi_j$.*

Sequentially, we provide that the Granger Causality in the estimated function $\mathbf{f}_j$ is identifiable. Proof of the theoretical results are shown in Appendix A2.

**Proposition 1** *(Identification of Granger Causality in the estimated function $\mathbf{f}_j$.) Suppose that the estimated function $\mathbf{f}_j$ well models the relationship between $\mathbf{x}_i$ and $\mathbf{x}_j$ from the training data. Given*

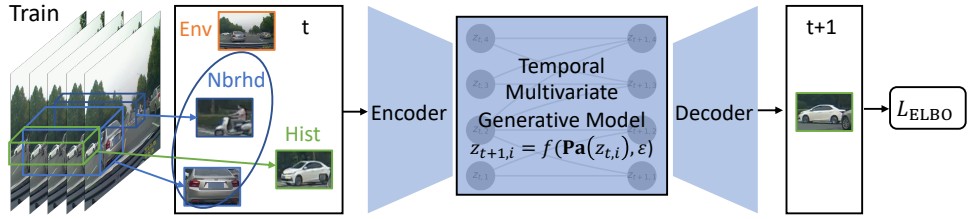

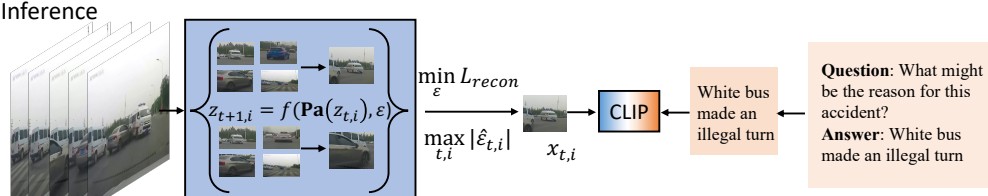

Figure 4: **Overall framework of LLCP.** In the training, we first obtain agents from regular videos by a tracking model, then apply these agents to fit a temporal multivariate generative model to learn the causal processes. Note that "Hist", "Nbrhd", "Env" denote history, neighborhood, and environment, respectively. This model is learned by the ELBO loss with self-supervision. In the inference, we first apply the learned generative model to process the test data, and then identify the cause variable (changing factor) which doesn't follow the regular causal processes. With identified cause variables and language keywords parsed from the reasoning task, we can give the answer by matching them with pre-trained CLIP models.

*the ground truth Granger Causal structure $\mathcal{G}(V, E_V)$ with the maximum time lag of 1, where $V$ and $E_V$ denote the nodes and edges, respectively. Assume that the data are generated by Equation (1), then $\mathbf{x}_i \to \mathbf{x}_j \notin E_V$ if and only if $\frac{\partial \mathbf{x}_{t,j}}{\partial \mathbf{x}_{t-1,i}} = 0$.*

**Network.** Given the features of observed variables extracted from a frozen CLIP image encoder, we model TMGM as an extension of Conditional Variational Auto-Encoders Sohn et al. (2015) (CVAE) with tailored modules to enforce the causal processes as Eq. 1. As shown in Figure 3, TMGM consists of three modules including a learnable encoder network to map the features of observed variables into the latent space, a decoder to generate the observed variables with the latent variables, and a transition module modeling the latent causal dynamics. In the transition module, we respectively learn three sub-networks $f_N$, $f_H$, and $f_E$, to learn the transitions from historical states (green), neighborhood interaction (blue), and environment information (orange). Then, we apply another sub-network to fuse the information that comes from different sources. All sub-networks are built by the Multilayer Perceptron (MLP).

**Optimization.** To train the TMGM, we formulate it as an auto-regression task that predicts the future state with past information. Specifically, it is optimized by the classical ELBO objective:

$$\mathcal{L}_{\text{ELBO}} = \sum_{t=1}^{T} \sum_{i=1}^{N} \mathcal{L}_{\text{Recon}}(\mathbf{x}_{t,i}, \hat{\mathbf{x}}_{t,i}) - \beta \mathbb{E}_{\hat{\mathbf{z}}_{t,i} \sim q} \log q\left(\hat{\mathbf{z}}_{t,i} | \mathbf{x}_{t,i}, \mathbf{Pa}(\hat{\mathbf{z}}_{t,i})\right) - \log p(\hat{\mathbf{z}}_{t,i} | \mathbf{Pa}(\hat{\mathbf{z}}_{t,i})), \quad (3)$$

where the $\mathcal{L}_{\text{Recon}}$ to calculate the distance between the generation and the ground truth is implemented by the binary cross-entropy function, and $\beta$ is a hyperparameter to balance two loss terms. The KL divergence $\mathcal{L}_{\text{KLD}}$ between the prior distribution $p(\hat{\mathbf{z}}_{t,i} | \mathbf{Pa}(\hat{\mathbf{z}}_{t,i}))$ and posterior distribution $q\left(\hat{\mathbf{z}}_{t,i} | \mathbf{x}_{t,i}, \mathbf{Pa}(\hat{\mathbf{z}}_{t,i})\right)$. The implementation details can be found in Appendix A4.

## 2.2 Identifying Root Cause of Accident

**Definition of Root Cause.** In a temporal dynamic system, the root causes of accidents are defined as the variables that significantly diverge from the established local causal mechanisms Sheet (2016); Yang et al. (2022b). Basically, these alterations occur exclusively from the local causal mechanism, unaffected by other extraneous variables. To illustrate, consider a traffic accident scenario where the

root causes can clearly be attributed to individuals or vehicles that disregard traffic rules, rather than the agents impacted by these variables.

**Implementation.** We apply the TMGM method to learn the stationary local causal mechanisms of each variable, by analyzing transitions from historical states to the present states under normal events. When a variable serves as the root cause, a notable shift in its local causal mechanisms becomes evident. To identify the changing factors, we contrast the observed transitions of each variable in the query sample against the causal mechanisms established through TMGM. The variable demonstrating deviation from the learned transitions potentially stands under the influence of altering factors. Specifically, we propose two criteria to examine whether the query variable is the root cause, including the Maximum Reconstruction Error (MRE) and the Outlier Reconstruction Noise (ORN).

**Maximum Reconstruction Error.** MRE serves as a direct method to confirm whether a variable aligns with the local causal mechanisms by analyzing the reconstruction error. Typically, variables exhibiting a larger reconstruction error are more likely to contain changing factors. Given a query video, we first detect the variables $\mathbf{X}' = \{\mathbf{x}'_{t,i}|_{t=1,i=1}^{T,N}\}$ in the same manner as the training stage. For each variable $\mathbf{x}'_{t,i}$, we can generate its predictions with our learned TMGM model using its parent variables. The process for detecting the root cause variable using MRE is as follows:

$$\mathbf{x}'_C = \arg\max_{t,i} ||\hat{\mathbf{x}}'_{t,i} - \mathbf{x}'_{t,i}||, \tag{4}$$

where $\hat{\mathbf{x}}'_{t,i}$ is the expected prediction obtained by the learned TMGM model $\mathbf{f}\left(\{\mathbf{z}_{t-\tau,j}|\mathbf{z}_{t-\tau,j} \in \mathbf{Pa}(\mathbf{z}_{t,i})\}, \mathbf{0}\right)$.

**Outlier Reconstruction Noise.** An alternative manner is to detect outliers by analyzing noise variables–with a fixed functional form of causal influence, we consider the instance as an outlier if the value of the noise is an outlier. Specifically, we generate a set of predictions $\hat{\mathbf{x}}'_{t,i}(\epsilon_k)$ with different samples $\epsilon_k$ of the noise variable, and find the noise value with the best prediction (the one with minimal predictive errors) as

$$\hat{\epsilon}_{t,i} = \arg\min_{\epsilon_k} ||\mathbf{x}'_{t,i} - \hat{\mathbf{x}}'_{t,i}(\epsilon_k)||. \tag{5}$$

Given the prior distribution of noise variables can be touched in Eq.1, i.e., a standard Gaussian distribution $\mathcal{N}(\mathbf{0}, \mathbf{I})$, we can conduct the statistical test to verify whether (or how) $\hat{\epsilon}_{t,i}$ follows the prior distribution. Variables that demonstrate notable deviations in best-reconstruction noise from prior expectations are identified as root causes.

## 2.3 Counterfactual Prediction

**Definition.** The counterfactual prediction in the causal dynamic system aims to reason the potential development if some historical actions are changed. Mathematically, let $\mathbf{Z}_H$ and $\mathbf{Z}_F$ denote the sets of history and future latent variables, respectively. The counterfactual prediction $\mathbf{Z}_F(\mathbf{u}|\mathbf{Z}_H = \mathbf{z}_h) = \mathbf{z}_f$ denotes that "$\mathbf{Z}_F$ would be $\mathbf{z}_f$, given the situation $\mathbf{u}$, if had $\mathbf{Z}_H$ been $\mathbf{z}_h$".

**Implementation.** Given the factual evidence (existing event $\mathbf{X}$), this counterfactual prediction $\mathbf{Z}_F(\mathbf{u}|\mathbf{Z}_H = \mathbf{z}_h)$ can be estimated with the following steps Pearl (2009):

- **Abduction**: Estimate the situation $\mathbf{u}$ using existing factual event $\mathbf{X}$ as $P(\mathbf{u}|\mathbf{X})$. We employ the learned TMGM model given by $\mathbf{f}\left(\{\mathbf{z}_{t-\tau,j}|\mathbf{z}_{t-\tau,j} \in \mathbf{Pa}(\mathbf{z}_{t,i})\}, \epsilon\right)$ to represent the local causal mechanism. The variable $\epsilon$ represents the situation variable that requires estimation. For a given query video $\mathbf{X}'$, we initially produce a set of predictions by introducing different noises, represented as $\hat{\mathbf{x}}'_{t,i}(\epsilon_k)$. We then select the prediction with the optimal reconstruction as the estimated $\hat{\mathbf{u}}$, as outlined in Equation (5).

- **Action**: Take the action $\text{do}(\mathbf{Z}_H = \mathbf{z}_h)$ to modify the causal process into counterfactual world. Based on the textual description of the counterfactual condition, we derive the textual embedding through an encoder, treating it as the estimated visual counterfactual condition $\hat{\mathbf{z}}_h$.

- **Prediction**: Use the modified causal process to predict counterfactuals as $\mathbf{Z}_F(\mathbf{u}|\mathbf{Z}_H = \mathbf{z}_h)$. With the estimated situation $\hat{\mathbf{u}}$ and counterfactual condition $\hat{\mathbf{z}}_h$ at hand, we adjust the relevant variables in the TMGM model to forecast possible effects. For these counterfactual conditions, we modify only the changing variables, retaining the rest in their original state.

## 2.4 ANSWER REASONING QUESTION

In this section, we demonstrate how the derived causal/effect variables can be applied to effectively answer fundamental reasoning questions. Prior to this, we first outline a fundamental characteristic of reasoning questions that allows them to be differentiated from reasoning-free questions. At their core, reasoning questions necessitate understanding variable "**changes**" and the corresponding impacts. To clarify, we provide several representative examples of reasoning questions, drawing attention to the notion of variable "changes". Consider the question $Q1$: "*What is the reason for the accident?*". It could be rephrased as: "*Which variable **change** led to the accident?*". Similarly, the question $Q2$: "*What would happen if the rope broke?*", translates to: "*What would occur if the rope's status **changes** from intact to broken?*". Lastly, $Q3$: "*Would the accident still happen if the blue car did not run a red light?*", can be understood as: "*Would the accident still happen when the state of variable 'blue car' is **changed** to 'did not run a red light'?*".

This suggests that the reasoning questions center on the variable "changes", evident in both questions ($Q2$, $Q3$) and answers ($Q1$). Furthermore, these questions can be segmented into two types: inferring causes ($Q1$, $Q3$) and predicting counterfactual effects $Q2$. Specifically, $Q1$ and $Q2$ represent attribution and counterfactual prediction respectively. For $Q3$, though it also follows the sentence pattern of $Q2$ such as " ... happen if ...", it is a true-or-false question but not an open-choice question. Such a question can be answered via attribution reasoning. For instance, if the established cause is not 'run a red light', the answer to this question is "no".

Technically, for open-choice attribution questions like $Q1$, we measure the distance between the identified root cause and answer candidates, choosing the most fitting one as the answer. This distance calculation can be facilitated by a pretrained vision-language model, such as CLIP Radford et al. (2021). In the case of open-choice counterfactual prediction questions like $Q2$, we begin by estimating potential counterfactual effects with counterfactual inference using the trained model, followed by comparing the effect to textual candidates. For true-or-false questions like $Q3$, we initially identify the key object of the question, such as 'blue car'. It is accomplished via language parsing, particularly by examining conditional adverbial clauses. (Additional parsing outcomes are available in the Appendix A6.4.) We then determine if this subject aligns with the identified root causal variable by gauging its cross-matching similarity against a set threshold. This match indicates that modifying this element might prevent the incident (answer yes) or not (answer no).

## 3 EXPERIMENTS

We executed the experiments in two synthetic and two real-world VideoQA datasets. In the simulation tests, we assess the capability of **LLCP** to identify the root cause and predict counterfactual effects, utilizing the ground truth as a benchmark. For the real-world tests, we evaluate **LLCP** with reasoning-based VideoQA tasks related to cause identification and counterfactual prediction, respectively.

### 3.1 SIMULATION EXPERIMENTS

In this section, we conducted these types of simulation experiments to evaluate whether **LLCP** can learn the causal process and how it can help downstream inferences. The first one focuses on root cause identification, which aims to attribute the cause of accidents. The second one is counterfactual inference, predicting the possible state with the given factual evidence and counterfactual conditions. We further conduct simulation experiments to demonstrate the proposed LLCP can uncover underlying causal relationships, whose details are shown in Appendix A5.3.

**Root cause identification.** In this experiment, we examine two potential root causes: function change (where the accident occurs during the generation process) and structure change (where the accident pertains to the causal structure). We construct a random causal structure, $G$, comprising 10 nodes, and introduce both function and structure changes to simulate accidents. Our dataset comprises a training set of 700 regular samples, and a test set of 1243 regular samples and 543 accident samples. For assessment, we employ Recall and F1-score as our metrics, marking an evaluation as false if either metric is not met.

Table 1 displays the experimental results. We have considered both VAE and LSTM-VAE as baseline methods. For each method, we employed five random seeds and reported the average outcomes.

Additionally, we conducted the Wilcoxon signed-rank test on these results. **LLCP** distinctly surpasses the baselines, with a p-value threshold set at 0.05. Some observations are summarized below. Firstly, in terms of recall performance, the proposed approach markedly excels over other methods across all datasets, indicating **LLCP** 's effectiveness in accurately identifying true causal variables with minimal misses. Furthermore, **LLCP** also demonstrates considerable advantages in the F1-Score metric, signifying its efficiency in minimizing false detections—particularly crucial in scenarios with sparse accident samples. Delve deeper into data generation specifics and extended experimental analysis in Appendix A5.

**Counterfactual inference.** For simulating counterfactual inference, we generate paired validation data comprising both factual evidence and counterfactual events (which share identical noise). During the inference phase, we examine the factual evidence and forecast the counterfactual effects under counterfactual conditions, using the trained model. Our dataset consists of 32,000 training samples and 16,800 paired samples (both factual and counterfactual) for evaluation. The RMSE of the prediction serves as our evaluation metric.

Table 1: Performance comparison (Recall and F1 Scores) on the simulation datasets.

| Methods | Function change | | Structure change | |
|---|---|---|---|---|
| | Recall | F1 Score | Recall | F1 Score |
| LSTM-VAE | 52.11 | 27.81 | 37.75 | 20.78 |
| VAE | 51.92 | 27.33 | 31.08 | 19.35 |
| **LLCP** | **63.34** | **63.01** | **45.04** | **44.19** |

Table 2 displays the experimental results. Predictions are evaluated under conditions with and without counterfactual inference. The tables provided are averages derived from three distinct experiments. Furthermore, after employing the Wilcoxon signed-rank test on these findings, it's evident our approach markedly surpasses the baselines, given a p-value threshold set at 0.05. It's worth emphasizing that the model that incorporates counterfactual inference consistently demonstrates superior performance compared to its counterpart. Further elaboration can be found in Appendix A5.

Table 2: Performance comparison (RMSE) on the simulation datasets.

| w/ Counterfactual Inference | w/o Counterfactual Inference |
|---|---|
| **0.0936** | 0.0967 |

## 3.2 REAL-WORLD EXPERIMENTS: SUTD-TRAFFICQA

We conduct experiments on the SUTD-TrafficQA Xu et al. (2021) dataset to evaluate the efficacy of **LLCP** in terms of traffic accident reasoning. The questions in SUTD-TrafficQA focus on the analysis of accidents, i.e., identifying the root cause of accidents.

**Dataset and experimental settings.** SUTD-TrafficQA consists of 10,080 traffic video sequences depicting various scenarios, along with over 60,000 QA sets, where 56,460 QA sets are used for training and the rest 6,075 are used for testing. It includes 6 tasks, including "Basic Understanding", "Event Forecasting", "Reverse Reasoning", "Counterfactual Inference", "Introspection", and "Attribution". For our study, we primarily focus on causality-based question types, such as "Attribution", "Introspection", and "Counterfactual", as prediction-based problems can be solved using conventional likelihood-based methods. Specifically, we selected approximately 2,000 questions for our experiments. To ensure that the questions were challenging and causal-related, we removed questions that focused on basic prediction and videos without any tracked variables. For a detailed list of the selected questions, please refer

Table 3: **More Ablation study on SUTD-TrafficQA.** We focus on the reasoning tasks including **C**: "Counterfactual inference", **I**: "Introspection", and **A**: "Attribution". CLIP* denotes using the template to refine answer candidates in the unsupervised way as Chen et al. (2023).

| Methods | SUTD-TrafficQA | | | |
|---|---|---|---|---|
| | C | I | A | Avg |
| CLIP Radford et al. (2021) | 32.6 | 23.5 | 27.7 | 27.7 |
| CLIP* Chen et al. (2023) | 43.4 | 34.8 | 30.1 | 31.2 |
| w/o $f_N$ | 49.7 | 45.6 | 30.1 | 31.4 |
| w/o $f_E$ | 48.8 | 46.1 | 30.4 | 31.8 |
| random $f_N$ | 49.1 | 44.6 | 30.0 | 32.2 |
| random $f_E$ | 45.6 | 45.6 | 30.2 | 32.2 |
| w/ Order | 47.8 | 47.3 | 30.7 | 32.4 |
| **LLCP** (ORN) | 49.6 | **48.2** | 30.8 | 33.2 |
| **LLCP** (MRE) | **50.4** | 46.4 | **31.3** | **33.5** |

to the supplementary materials. All tasks were formulated as multiple-choice questions, with no limitations on the number of candidate answers. The evaluation metric we used was the accuracy of the multiple-choice responses.

**Ablation study.** Here we compare **LLCP** with the baseline and other ablation versions on SUTD-TrafficQA. The ablation versions consist of "w/o $f_N$", "w/o $f_E$", and "w/ Order", where "w/o $f_N$" and "w/o $f_E$" denote that we remove the $f_N$ or $f_E$ part. "w/ Order" denotes that we don't use the shared network for different neighborhoods but build sub-network for each of them, where we use the spatial distance to calculate the order of neighborhoods to avoid misalignment.

Table 3 summarizes the performance of all versions of **LLCP** and the baseline unsupervised CLIP Radford et al. (2021) method. We can draw the following conclusions from the comparison. First, comparing the CLIP baseline with our **LLCP** , we observe that learning causal process with a multivariate generative model can obtain overall significant improvement. Second, removing $f_N$ and $f_E$ hurts the performance, which demonstrates both these two modules help learn the causal generation process. Beyond "w/o $f_N$", "w/o $f_E$", we added "random $f_N$" and "random $f_E$". We found that the parameters also influence the performance since the performance of using the random feature is a bit better than the total removal. Second, using random $f_N$ and $f_E$ hurts the performance, which demonstrates both these two modules help learn the causal generation process. Third, when we use different sub-networks for different neighborhoods and introduce the order, the performance drops. It is reasonable since the calculation of orders might be inaccurate since the spatial distance in the image is affected by 3D-to-2D projection and the quality of tracking models. Besides, we also compare two kinds of methods (ORN and MRE) to identify the cause variable. We find that both of them achieve good performance and **LLCP** (MRE) is a little higher.

**Compared with supervised SOTAs.** We also compare our **LLCP** with other state-of-the-art **supervised** methods. The results are summarized in Table 4. We apply **LLCP** to reasoning-based questions and employ the unsupervised CLIP model for the other reasoning-free questions, which directly matches the video and the declarative version of QA pairs. Though without using the QA annotation (only using whether the video is regular or not), our **LLCP** achieves promising results compared with other supervised methods. Specifically, we outperform some baseline methods such as CNN+LSTM, I3D+LSTM, and VIS+LSTM Ren et al. (2015a). Besides, compared to the recent method ATP Buch et al. (2022), which also used the CLIP and further

Table 4: **Comparison with the state-of-the-art methods on SUTD-TrafficQA. QA** denotes whether the QA pair annotation is used for training.

| Methods | QA | Acc |
|---|---|---|
| Avgpooling | | 30.5 |
| CNN+LSTM | | 30.8 |
| I3D+LSTM | | 33.2 |
| VIS+LSTM Ren et al. (2015a) | | 29.9 |
| BERT-VQA Yang et al. (2020) | Yes | 33.7 |
| TVQA Lei et al. (2018) | | 35.2 |
| HCRN Le et al. (2020) | | 36.5 |
| Eclipse Xu et al. (2021) | | 37.1 |
| Tem-Adapter Chen et al. (2023) | | 46.0 |
| ATP + CLIP Buch et al. (2022) | | 35.6 |
| CLIP Radford et al. (2021) | No | 27.5 |
| **LLCP** + CLIP | | 33.3 |

learned an adapter module with the annotated supervision signals, **LLCP** can still achieve comparable performance.

## 3.3 REAL-WORLD EXPERIMENTS: CAUSAL-VIDQA

To further assess the capability of **LLCP** for counterfactual prediction tasks, we have performed experiments on the Causal-VidQA Li et al. (2022a) dataset. We have chosen not to include the NextQA dataset in this study as they possess identical question types.

**Dataset and experimental settings.** Questions in the Causal-VidQA dataset are divided into four types: description, explanatory, prediction, and counterfactual. In this study, we only focus on reasoning-based questions, specifically those of counterfactuals. We conduct evaluations on both validation and test sets. We follow the experimental protocol in Li et al. (2022a) to report the results.

**Results and Analysis.** As shown in Table 5, we compared **LLCP** with other state-of-the-art methods. Every method was re-implemented using the official code, uniformly employing the same CLIP feature. Compared with the CLIP baseline, **LLCP** registers an improvement of nearly 10 percent on average. Remarkably, even when compared against other supervised learning approaches, our method yields results that are comparable, despite the absence of annotated data for training.

Table 5: **Comparison with baseline methods on Causal-VidQA.** We re-train all methods using the same CLIP features using the officially released code and report the results on both validation and test datasets related to reasoning tasks. More results are in the Appendix.

| Methods | QA | Validation_C | | | Test_C | | | Avg |
|---|---|---|---|---|---|---|---|---|
| | | Q → A | Q → R | Q → AR | Q → A | Q → R | Q → AR | |
| EVQA Antol et al. (2015) | Yes | 28.27 | 28.05 | 9.54 | 28.05 | 28.05 | 10.09 | 22.00 |
| CoMem Gao et al. (2018) | | 47.61 | 47.01 | 25.60 | 45.70 | 47.60 | 25.44 | 39.83 |
| HME Fan et al. (2019) | | 46.09 | 47.12 | 25.53 | 45.48 | 46.51 | 24.74 | 39.25 |
| HCRN Le et al. (2020) | | 45.64 | 46.01 | 24.79 | 44.26 | 45.64 | 24.35 | 38.49 |
| HGA Jiang & Han (2020) | | 45.57 | 45.75 | 24.38 | 45.28 | 46.80 | 24.81 | 38.77 |
| B2A Park et al. (2021) | | 48.83 | 48.98 | 27.68 | 47.41 | 48.74 | 27.39 | 41.51 |
| CLIP Radford et al. (2021) | No | 31.61 | 28.50 | 11.32 | 29.95 | 29.51 | 11.48 | 23.73 |
| **LLCP** | | **38.36** | **38.91** | **17.93** | **39.07** | **38.46** | **19.03** | **32.13** |

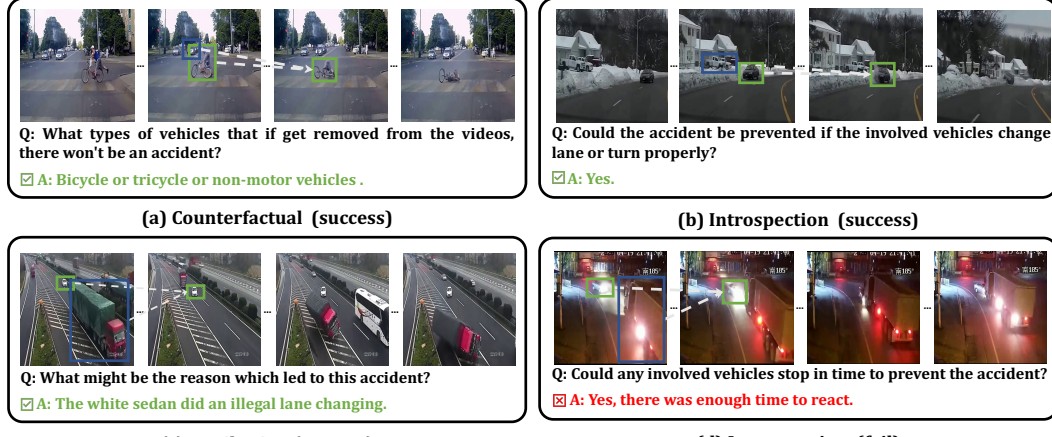

Figure 5: **Visualization results of LLCP on SUTD-TrafficQA.** We show 3 positive examples of different question types including (a) Counterfactual, (b) Introspection, and (c) Attribution. Besides, we also provide a failure case (d) Introspection.

## 3.4 QUALITATIVE RESULTS AND DISCUSSION

In this section, we provide some qualitative results on the SUTD-TrafficQA dataset in Figure 5, to analyze our framework by case. We provide both success and failure cases on different question types, such as (a) Counterfactual, (b) Introspection, and (c) Attribution, to understand the methods and explore the boundary. We provide the question, our answer, and our learned causal process to show why **LLCP** can work or fail. Taking (a) Counterfactual as an example, **LLCP** identifies the root cause as "bicycle", i.e., the bicycle doesn't follow the normal causal mechanism (fall down). For (c) Attribution, **LLCP** can identify the abnormal action of the white car (the illegal lane changing) given its neighborhoods and environmental information. For the failure case (d), though the model can identify the cause variable, the understanding of the current variable's state is incorrect, which leads to the failure.

## 4 CONCLUSION

In this paper, we introduce a novel framework, **LLCP**, designed to tackle reasoning-based VideoQA through a causality lens. In this framework, we exploit the modularity of causal mechanisms to develop a temporal multivariate generative model tailored for local causal processes. During the inference phase, **LLCP** can attribute the accidents by identifying the root cause, or forecast the potential outcomes under counterfactual scenarios. Furthermore, aligning the visual and textual domains with CLIP allows us to respond to queries using neural language. In the experiments, our framework has proven effective on both synthetic and real-world datasets.

ACKNOWLEDGEMENT

This material is based upon work supported by the AI Research Institutes Program funded by the National Science Foundation under AI Institute for Societal Decision Making (AI-SDM), Award No. 2229881. The project is also partially supported by the National Institutes of Health (NIH) under Contract R01HL159805, and grants from Apple Inc., KDDI Research Inc., Quris AI, and Infinite Brain Technology.

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

*Appendix for*

**"LLCP: Learning Latent Causal Processes for Reasoning-based Video Question Answer"**

Appendix organization:

Table A1: A summary and classification of VideoQA datasets.

| Datasets | MSVD-QA Xu et al. (2016) | MSRVTT-QA Xu et al. (2016) | ActivityNet-QA Yu et al. (2019) | TVQA Lei et al. (2018) | MovieQA Tapaswi et al. (2016) | CLEVRER Yi et al. (2020) | NExT-QA Xiao et al. (2021) | TrafficQA Xu et al. (2021) | Causal-VidQA Li et al. (2022a) |
|---|---|---|---|---|---|---|---|---|---|
| Reasoning | ✗ | ✗ | ✗ | ✗ | ✗ | ✓ | ✓ | ✓ | ✓ |
| Multimodal | ✗ | ✗ | ✗ | ✓ | ✓ | ✗ | ✗ | ✗ | ✗ |
| Real Videos | ✓ | ✓ | ✓ | ✓ | ✓ | ✗ | ✓ | ✓ | ✓ |

## A1 RELATED WORKS

**Video Question Answering.** Depending upon the nature of the questions, VideoQA tasks can be categorized into two distinct sectors: perception-based VideoQA and reasoning-based VideoQA, as shown in Table A1. The former, as evidenced in studies such as Xu et al. (2016); Yu et al. (2019); Tapaswi et al. (2016); Lei et al. (2018; 2019), focuses on deciphering video content to respond to natural language inquiries concerning visual facts such as location (where), time (when), object (who), attributes (e.g., color or quantity), and manner (how). A core hurdle here is achieving seamless cross-modality matching. On the other hand, reasoning-based VideoQA, documented in Xu et al. (2021); Yi et al. (2020); Li et al. (2022a); Xiao et al. (2021), concentrates on analyzing temporal dynamics and discerning the causality of video events, addressing 'why' or 'what if' types of questions. This necessitates the model's comprehension of the causal relations among the agents. Conventional methods for reasoning-based VideoQA often delineate the task through the lens of cross-modality matching Ji et al. (2020); Cong et al. (2021); Feng et al. (2021); Le et al. (2020); Gao et al. (2018); Lei et al. (2021); Bain et al. (2021); Yang et al. (2022a). Yet, these strategies are heavily dependent on extensive paired annotated data, and may be misled by spurious correlations present in the training sets. Recently, several methods Wei et al. (2023); Zang et al. (2023); Li et al. (2022b) have incorporated causal inference techniques to mitigate bias and enhance resilience. However, these approaches necessitate prior understanding of established causal frameworks, alongside paired annotations. In this study, we propose a groundbreaking framework, `LLCP`, which eliminates the need for paired data and prior knowledge of causal structures.

**Abnormal Event Detection.** Abnormal event detection methods often learn a familiarity model to distinguish the outlier samples from the pool of regular videos. To build the regular distribution, existing methods apply dictionary learning methods Cong et al. (2011); Lu et al. (2013); Cheng et al. (2015) or deep neural networks including deep generic knowledge Hinami et al. (2017), stacked RNN Luo et al. (2017), C3D Sabokrou et al. (2017), and plug-and-play CNNs Hasan et al. (2016); Liu et al. (2018). Unlike abnormal event detection which only focuses on the distribution of the whole event, we go further to understand the underlying causal processes of the event and find the reason why this event is abnormal.

**Causal Representation Learning.** Recently, causal representation learning methods have demonstrated remarkable success in identifying the latent variables from time-series data by learning the causal latent processes in an unsupervised manner Yao et al. (2022b;a). Unlike these methods of identifying latent variables, in this paper, we focus more on how to apply the learned causal mechanism to discover changed factors and analyze the video event.

## A2 THEORETICAL ANALYSIS

In this section, we will theoretically analyze the identification of the proposed model. To achieve this, we begin with the definition of the Granger non-causality of multi-type object sequences as follows.

**Definition 1** (*Granger non-causality of multi-type object sequences*) *Suppose that the object sequences* $\mathbf{X}$ *are generated according to Equation 1, we can determine the Granger non-causality of the object* $\mathbf{x}_i$ *with respect to object* $\mathbf{x}_j$ *as follows: For all* $\mathbf{x}_{t-1:t-\tau,1}, \mathbf{x}_{t-1:t-\tau,2}, \cdots, \mathbf{x}_{t-1:t-\tau,N}$, *and the same variable with different values* $\mathbf{x}_{t-1:t-\tau,i} \neq \mathbf{x}'_{t-1:t-\tau,i}$, *if the following condition holds:*

$$\phi_j(\mathbf{x}_{t-1:t-\tau,1}, \cdots, \mathbf{x}_{t-1:t-\tau,i}, \cdots, \mathbf{x}_{t-1:t-\tau,N}) = \phi_j(\mathbf{x}_{t-1:t-\tau,1}, \cdots, \mathbf{x}'_{t-1:t-\tau,i}, \cdots, \mathbf{x}_{t-1:t-\tau,N}),$$

$$(6)$$

*that is,* $\mathbf{x}_{t,j}$ *is invariant to* $\mathbf{x}_{t-1:t-\tau,i}$ *with* $\phi_j$.

**Proposition 1** (*Identification of Granger Causality in the estimated function* $\mathbf{f}_j$.) *Suppose that the estimated function* $f_j$ *well models the relationship between* $\mathbf{x}_i$ *and* $\mathbf{x}_j$ *from the training data. Given the ground truth Granger Causal structure* $\mathcal{G}(V, E_V)$ *with the maximum time lag of 1, where* $V$ *and*

$E_V$ denote the nodes and edges, respectively. Assume that the data are generated by Equation (1), then $\mathbf{x}_i \rightarrow \mathbf{x}_j \notin E_V$ if and only if $\frac{\partial \mathbf{x}_{t,j}}{\partial \mathbf{x}_{t-1,i}} = 0$.

**Proof 1** $\Longrightarrow$: If $\mathbf{x}_i \rightarrow \mathbf{x}_j \notin E_V$ (there is no Granger Causality between $\mathbf{x}_i$ and $\mathbf{x}_j$ in the ground truth process), then according to Definition 1, there must be $\mathbf{x}_j = \mathbf{x}'_j$ for all the different values of $\mathbf{x}_i$. If $\frac{\partial \mathbf{x}_{t,j}}{\partial \mathbf{x}_{t-1,i}} = 0$, then when the input values are different, i.e. $\mathbf{x}_i \neq \mathbf{x}'_i$, the outputs of $\mathbf{f}_j$ are also different, i.e., $\mathbf{x}_j \neq \mathbf{x}'_j$, which results in contradictions.

$\Longleftarrow$: Suppose $\mathbf{x}_i \rightarrow \mathbf{x}_j \in E_V$ (there is Granger Causality between $\mathbf{x}_i$ and $\mathbf{x}_j$ in the ground truth process), for any $\mathbf{x}_j$ and $\mathbf{x}'_j$ in Definition 1, if $\mathbf{x}_j \neq \mathbf{x}'_j$, there must be different input $\mathbf{x}_i \neq \mathbf{x}'_i$. If $\frac{\partial \mathbf{x}_{t,j}}{\partial \mathbf{x}_{t-1,i}} = 0$, then when $\mathbf{x}_j \neq \mathbf{x}'_j$, there might be $\mathbf{x}_i = \mathbf{x}'_i$, which results in contradictions.

## A3 FURTHER ANALYSIS AND DISSUSIONS OF **LLCP**

### A3.1 ASSUMPTIONS OF **LLCP**

To well illustrate our method, we intuitively discuss the assumptions inherent in ***LLCP***, which are crucial for understanding the causality in the temporal dynamic system. To strengthen the credibility and rationality of our outlined assumptions, we also delve deeper into their practical applications and justifications:

1. **Granger Causality Principle:** It is assumed that the past values of one variable can predict the future values of another. This implies a directional relationship between variables in a time-series context. It also indicates there are no instantaneous effects in the temporal dynamics, i.e., the variables $\mathbf{z}_t$ are independent given the historical state $\mathbf{Pa}(\mathbf{z}_t)$. This assumption aligns seamlessly with the data generation process, as described in Eq. 1. **Implementation:** Implementationally, we employed auto-regression to capitalize on the concept of Granger Causality. This method predicates future events based on historical data, thereby validating our assumption about the predictability of one variable's future values based on another's past values. We provided a proposition and corresponding proof to show the identification of the proposed model in Section A2.

2. **Shared Feature Spaces in Visual and Textual Domains:** The assumption here is that the feature spaces for both visual and textual domains are shared, allowing for matching the same variables across these domains. **Implementation:** To validate this assumption about the shared feature spaces in visual and textual domains, we incorporated the pre-trained CLIP model. This model acts as a feature extractor for both video objects and textual elements (questions and answers). Its application ensures that both visual and textual data are represented in congruent feature spaces, thereby affirming the feasibility of matching variables across these domains.

### A3.2 LIMITATIONS

One notable limitation in our approach is the reliance on the Granger Causality assumption, which can be considered somewhat robust. For the sake of simplicity, our current model does not account for more complex scenarios, such as the presence of latent confounders or non-stationary conditions. These factors, while critical in certain contexts, are beyond the scope of our initial assumptions. Despite this, our application of Granger Causality serves as a foundation for a potential framework. This framework is instrumental in understanding and leveraging causal relationships within temporal dynamics for subsequent reasoning tasks. Looking ahead, we aim to explore the identifiability of causal variables under less stringent assumptions. This progression will enable a more nuanced understanding of causal relationships in diverse and complex environments.

Another aspect of our methodology is its reliance on pre-trained models, notably CLIP and tracking models. The primary motivation behind employing these models is to minimize the necessity of textual annotations. We leave it as the future work to remove these assumptions and learn the latent variable from scratch with textual annotations.

### A3.3 RATIONALITY AND MOTIVATION FOR TRAINING WITHOUT TEXTUAL ANNOTATIONS

Please note that the usages of textual information in training and inference are different. We provide the discussions on usage proposals in both stages below.

**Training Without Textual Pairs.** It would be really good if we could always have enough annotated question-answering pairs as the supervision. It would help the learning of the reasoning model. However. the annotated question-answering pairs are not easy to obtain, especially for tasks requiring domain knowledge such as traffic accident attribution. For humans, learning to reason can be effectively achieved without cross-modality supervision, relying solely on observation induction within shared feature spaces. This concept aligns with causality theories Yao et al. (2022b;a), suggesting the feasibility of uncovering latent variables and causal relationships unsupervised. This approach is particularly relevant in real-world applications for several reasons. Firstly, the acquisition of annotated textual pairs is more challenging and costly compared to collecting unannotated videos. Additionally, models trained with textual annotations are susceptible to language biases, where they might learn shortcuts based on linguistic patterns rather than understanding the content. For instance, if all questions in the training set beginning with "What is the reason" have similar answers, the model might generalize inaccurately, associating any question with this format to a specific answer, such as "White car speeding", regardless of context. This phenomenon, known as language bias, has been extensively discussed in question-answering research Yuan (2021); Kv & Mittal (2020); Wen et al. (2021).

**Use of Text Queries in the Inference Phase.** Despite the avoidance of textual pairs in training, text queries play a crucial role during the inference phase. Our model, trained exclusively on visual data, can identify the visual causal relations of variables. During inference, textual queries are employed to first identify the local causal mechanisms we are interested in. Then, we leverage the causal relations to answer the detailed question. For example, if two accidents occur in the same video, our model will find causal mechanisms with two root causes. Then the query questions can help to identify the root cause we are more interested in.

### A4 IMPLEMENTATION DETAILS OF **LLCP**

In this section, we provide the implementation details of our approach for better reproduction.

**Models and Network Architecture.** In our experiments, we track the variables with QDTrack Pang et al. (2021) pre-trained on the TAO Dave et al. (2020) dataset. We select only traffic-related categories, such as "bicycle", "bus", "car", "motorcycle," and so on. After obtaining the bounding box, we extract the feature of each agent by the ROI Ren et al. (2015b) pool from the CLIP Radford et al. (2021) (RN50) image encoder. For the environment variable, we use the feature of the whole image and thus do not need to build specific causal processes for environment variables. Given the dimension of features from CLIP is 1024, the encoder network we used for CVAE is Linear(1024, 256). Correspondingly, the decoder is Linear(256, 1024). For the sub-networks $f_H$, $f_N$, and $f_E$ to learn the causal process from historical states, neighborhood interactions, and environmental clues, are built with Linear(1024, 256), respectively. In the experiments for SUTD-TrafficQA, we set the number of neighborhoods as 2, and they share the same $f_N$. Then we connect 4 local features and use a Linear(1024, 16) for information fusion. The dimension of the generative noise is set as 10. In the inference, the features of textual keywords are extracted by a CLIP text encoder.

**Hyperparameters.** We follow the video frames sampling strategy with Le et al. (2020). We first uniformly sample 8 frames from the video sequence, then with these frames as the center, we select 4 frames around each center. Finally, we obtained 32 frames for each video. For the loss function, we set the $\beta = 0.5$ to balance the reconstruction and KLD.

**Optimization Details.** We apply Adam Kingma & Ba (2014) as the optimizer with the initial learning rate as 1e-4 and batch size as 16. The learning rate decay is set to 0.5 to reduce the learning rate to half in every 10 epochs. The number of total epochs is set as 50. The model is implemented in PyTorch 1.13.1 and trained on a single Tesla V100 GPU.

## A5 MORE DETAILS OF SIMULATION EXPERIMENTS

### A5.1 ROOT CAUSAL IDENTIFICATION

**Dataset and experimental settings.** To evaluate whether `LLCP` can learn the causal process and identify the cause variables, we design a series of simulation experiments based on a random causal structure with a given sample size and variable size. To simulate the ground-truth circumstance, we assume that the training dataset does not contain any accident videos and that the test dataset contains regular videos and accident videos. The simulated datasets are generated in the following three steps.

First, we randomly generate a causal structure $G$ with the number of 10 nodes for the training data. Specifically, we let $G_{i,j}$ be the value in the $i$-th row and $j$-th column of $G$. Therefore, if $G_{i,j} \neq 0$, the $j$-th variable becomes the direct cause of $i$-th variable, i.e., $z_{t-1,j} \rightarrow z_{t,i}$. Since the causal structure might be sparse in reality, we assume that each node at most has three causes. Second, we recursively generate the simulated data with the nonlinear function:

$$z_{t+1,i} = \text{Sigmoid}(\sum_{j=0}^{k} G_{i,j} \times z_{t,j} + E_t) + \epsilon_{t+1}, \quad (7)$$

$$x_{t+1,i} = \phi(z_{t+1,i}),$$

where z denotes the latent variables and x denotes the observed variables; $\phi(\cdot)$ is a linear transformation from z to x; $E_t = sin(t)$ denotes the environment variables and $\epsilon_{t+1}$ denotes the noise variables. Third, to simulate accidents, we consider two types of changes, i.e., the function change and the structure change, as shown in

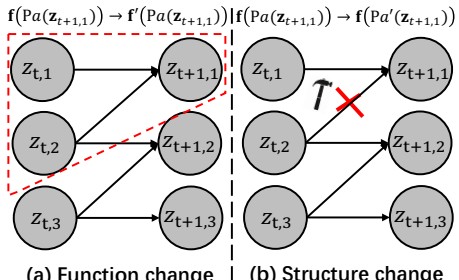

$\mathbf{f}(\text{Pa}(\mathbf{z}_{t+1,1})) \rightarrow \mathbf{f}'(\text{Pa}(\mathbf{z}_{t+1,1})) | \mathbf{f}(\text{Pa}(\mathbf{z}_{t+1,1})) \rightarrow \mathbf{f}(\text{Pa}'(\mathbf{z}_{t+1,1}))$

(a) Function change | (b) Structure change

Figure A1: **The examples of two types of change factors.** (a) Function change denotes that changes are caused by the generation function. For example, the generation function $\mathbf{f}(\text{Pa}(\mathbf{z}_{t+1,1}), \epsilon_{t+1,1})$ changes to $\mathbf{f}'(\text{Pa}(\mathbf{z}_{t+1,1}), \epsilon_{t+1,1})$. (b) shows the changes come from the causal structures, e.g. the edge $z_{t,2} \rightarrow z_{t+1,1}$ is blocked due to some accidents.

Figure A1. Function changes act on the nonlinear functions while structure changes act on the causal structure. The function changes are implemented as the direct value change of the latent variables:

$$z'_{t+1,i} = \alpha(z_{t+1,i} - \bar{z}_{1:t,i}) + \bar{z}_{1:t,i}, \quad (8)$$

in where $\bar{z}_{1:t,i}$ denotes the average value of $z_i$ from 1-st timestamp to $t$-th timestamp. For the structure change, we block some edges (e.g., $z_{t,2} \rightarrow z_{t+1,1}$) from $G$ and obtain intervened causal graph $G'$, and generate the accident value by replacing $G$ with $G'$ in Equation (7). Using these two types of accidents, we build a test set that contains 1243 regular samples and 543 accident samples.

**Evaluation metric.** Based on the aforementioned simulation dataset, we evaluate `LLCP` with two tasks including abnormal event detection to classify where the data contains accidents and changed factor identification to find the changed variable. We apply both Recall and F1-score as our evaluation metrics and we mark false when either one of them is wrong.

**Quantitative results.**

Experiment results on the simulation dataset are shown in Table A2, where the VAE, LSTM-VAE, Slow-VAE Klindt et al. (2020), and SKD Berman et al. (2023) are considered as compared to baseline methods. For all the methods, we try five different random seeds and report the average results. We also conducted the Wilcoxon signed-rank test on the results, our method significantly outperforms the baselines, with a p-value threshold of 0.05.

Table A2: Performance comparison (Recall and F1 Scores) on the simulation datasets.

| Methods | Function change | | Structure change | |
|---|---|---|---|---|
| | Recall | F1 Score | Recall | F1 Score |
| LSTM-VAE | 52.11 | 27.81 | 37.75 | 20.78 |
| VAE | 51.92 | 27.33 | 31.08 | 19.35 |
| SlowVAE | 54.41 | 29.03 | 38.12 | 21.39 |
| SKD | 58.93 | 34.76 | 35.72 | 19.79 |
| **LLCP** | **63.34** | **63.01** | **45.04** | **44.19** |

According to the experiment results, we have the following observations. First, compared with the performance of recall, we find that the performance of the proposed approach outperforms the other

methods on all the datasets with a large gap, reflecting that **LLCP** can identify true cause variables with low missed detections. This is because our method leverages causal mechanisms to infer the accident with the local causal process instead of detecting the variables with the abnormal distribution. Moreover, we find **LLCP** also enjoy more advantages in the metric of F1-Score, reflecting that it obtains low false detections and thus works well for the situation where accident samples are rare. We found that even compared with the advanced SKD Berman et al. (2023) model, **LLCP** can achieve significant improvement.

### A5.2 COUNTERFACTUAL PREDICTION

**Dataset and experiment settings.**

To evaluate whether the proposed method can address the counterfactual inference problem, we devise the counterfactual prediction simulation experiment.

The generation processes of factual and counterfactual data are shown in Figure A2. As for the factual datasets, we randomly generate a causal structure $G$ with the number of 10 nodes. Specifically, we let $G_{i,j}$ be the value in the $i$-th row and $j$-column of $G$. Similar to the simulated data in the function change scenario in A5.1, we assume that each node at most has three causes and the factual simulated datasets are recursively generated via Equation (7). As for the counterfactual datasets, we employ the same data generation process but different historical variables, which is shown in Equation(9).

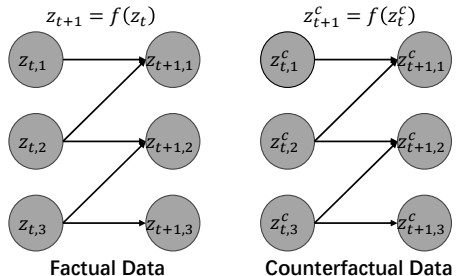

$$z_{t+1,i}^c = \text{Sigmoid}(\sum_{j=0}^{k} G_{i,j} \times (z_{t,j}^c + E_t) + \epsilon_{t+1},$$
$$x_{t+1,i}^c = \psi(z_{t+1,i}^c),$$

(9)

where $z^c$ denotes the counterfactual latent variables and $x^c$ denotes the observed counterfactual variables; $\psi$ is a nonlinear function. We use the historical observation $z_t$ as input and predict the future observation $z_{t+1}$. There are 32000 samples in the training dataset and 16800 pairs of factual and counterfactual samples in the test dataset for counterfactual inference.

Figure A2: **The examples of factual and counterfactual data.** Both factual and counterfactual data are generated via the same causal process. The counterfactual data are generated with perturbed input.

**Evaluation metric.** Based on the above simulation dataset, we employ Root Mean Square Error (RMSE) to evaluate our counterfactual prediction performance, the lower values of RMSE reflect a better performance of counterfactual inference.

**Quantitative result.** Experiment results on the simulation datasets are shown in Table A3. The w/o Counterfactual Inference denotes the model of standard variational autoencoder (VAE) and the w/ Counterfactual Inference denotes the VAE model with noise estimation like Pearl (2009). For all the methods, we try three different random seeds and report the average results. We also conducted the Wilcoxon signed-rank test on the results, our

Table A3: Performance comparison (RMSE) on the simulation datasets.

| w/ Counterfactual Inference | w/o Counterfactual Inference |
|---|---|
| **0.0936** | 0.0967 |

method significantly outperforms the baselines, with a p-value threshold of 0.05. According to the experiment results, we find that the model with noise estimation achieves a better performance than the normal VAE, showing that our method can address the counterfactual prediction problem.

### A5.3 GRANGER CAUSALITY DISCOVERY

To demonstrate the proposed LLCP can uncover underlying causal relations as opposed to merely correlational patterns, we further evaluate our method on the Granger Causality task. To achieve this, we follow the setting in Marcinkevičs & Vogt (2021) and evaluate the proposed methods on the simulated FMRI time series benchmark.

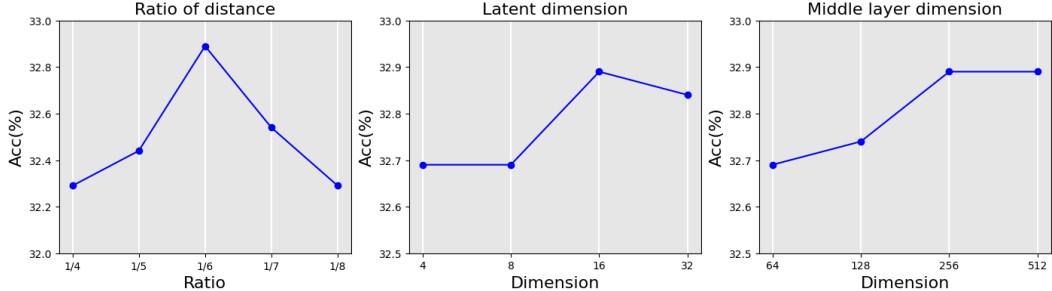

Figure A3: **Parameter analysis of LLCP.** We analyze three main hyperparameters including the distance ratio to fuse the local distance and global distance, the dimension of the latent space, and the dimension of the local sub-networks $(f_H, f_N, f_E)$, which show in the left, middle, and right, respectively.

To make our method generate the Granger Causality explicitly, we employ the conditional-VAE architecture and consider the Granger causal structures as the latent variables. As for the HCRN model, we obtain the estimated structure by an inner product between the extracted features from the HCRN model. We further consider four different metrics, i.e., accuracy (ACC),

Table A4: Experiments results on the capability to uncover Granger Causality.

| Models | ACC | BA | AUROC | AUPRC |
|---|---|---|---|---|
| LLCP | 0.788 | 0.632 | 0.657 | 0.289 |
| HCRN | 0.632 | 0.491 | 0.504 | 0.095 |

balanced accuracy (BA) scores, areas under receiver operating characteristic (AUROC), and precision-recall (AUPRC) curves. Experiment results are shown in Table A4. According to the experiment results, we can find that the proposed LLCP can capture the underlying causal relations.

## A6 MORE EXPERIMENTAL ANALYSIS

### A6.1 PARAMETERS ANALYSIS

In this section, we re-trained our model with different hyperparameters and provided the experimental results on the SUTD-TrafficQA dataset to investigate their effects. Specifically, we analyzed three main hyperparameters including the distance ratio to fuse the local distance and global distance, the dimension of the latent space, and the dimension of the local sub-networks $(f_H, f_N, f_E)$.

The local distance is calculated by the detected variable and language keywords and the global distance is calculated by the global visual features (the average of features of all frames) and textual description. We fuse these two complementary distances for better reasoning, where the local one can identify the separated causal variable while the global one focuses more on overall event perception. The left one of Figure A3 shows the results on the SUTD-TrafficQA dataset with different ratios. We found that our model achieves the optimal results with the ratio of $1/6$ between local and global distances.

As shown in the middle and right parts of Figure A3, we provide the performance of the models with different latent and middle layer dimensions. We observe that LLCP achieves better results when we increase the latent dimensions and middle layer dimensions, since the representation abilities increase with more parameters. However, we also find that this increase will converge when parameters are enough. Balancing the accuracy and cost, we set the latent dimension as 16 and the middle layer dimensions as 256, respectively.

### A6.2 COMPARED WITH LLMS-BASED METHODS

The LLMs-based methods Alayrac et al. (2022); Gan et al. (2022); Awadalla et al. (2023); Su et al. (2023) also don't require the reasoning QA pair for the training. To make a more complete comparison, we also compare our method with these LLMs-based methods, such as OpenFlamingo Awadalla et al. (2023). We evaluate all the models on the SUTD-TrafficQA dataset with a zero-shot setting. For

OpenFlamingo, we used the 2/4 textual demo examples (no visual data due to zero-shot) to guide the outputs. Here is an illustrative Python code of how to use the demo example in the prompt.

As shown in Table A5, we summarize the average accuracy of OpenFlamingo, CLIP, and LLCP. We found that the LLM models don't work as well as we expected since no particular knowledge is used. It is interesting and may inspire the following research.

```
[language=Python,
basicstyle=\ttfamily,
breaklines=true,
showstringspaces=false]
for i in range(len(questions)):
    token_inputs.append(
        "Question: What might have happened moments ago?
        Candidate 1: The blue truck hit the white sedan from the back.
        Candidate 2: The white sedan crashed into the blue truck.
        Candidate 3: The blue truck did an emergency brake.
        Candidate 4: The white sedan lost its control.
        Answer: The blue truck did an emergency brake.
        <|endofchunk|>
        Question: What might be the reason which led to this accident?
        Candidate1: The black sedan exceeded the speed limit
        Candidate2: The white sedan did an illegal lane changing
        Candidate3: Extreme weather condition
        Candidate4: The blue truck violated the traffic lights
        Answer: The white sedan did an illegal lane changing.
        <|endofchunk|>
        <image>
        Question:{} Candidate1:{} Candidate2:{} Candidate3:{} candidate4: {}
        Answer:".format(
            questions[i],
            ans_candidates[0][i],
            ans_candidates[1][i],
            ans_candidates[2][i],
            ans_candidates[3][i]
        )
    )
```

Table A5: Comparison with LLM-based methods on SUTD-TrafficQA

|  | OpenFlamingo (2 examples) | OpenFlamingo (4 examples) | CLIP | LLCP |
|---|---|---|---|---|
| Accuracy | 28.7 | 29.1 | 27.7 | 33.7 |

### A6.3 MORE EXPERIMENTAL RESULTS ON CAUSALVIDQA

We have included the results reported in the original CausalVidQA paper Li et al. (2022a). As detailed in table A6, we present the results of all baselines using the pre-trained 300-dimensional GloVe Pennington et al. (2014) word embeddings. It is evident that the CLIP features outperform the Glove features consistently. In our approach, we solely utilize the CLIP feature and eschew the Glove feature, owing to the pre-alignment of the visual and textual CLIP features. Notably, our method does not leverage annotated data and, thus, cannot learn cross-modality matching from scratch.

### A6.4 PARSING RESULTS

To explain our process of grammar parsing, which is used to extract key variables from the question-answers, we share detailed parsing results across varied question types. In Table A7, we enumerate the type of reasoning question, the original question itself, our logic rules employed for parsing, and the resulting keywords. Consider, for instance, an Open-choice Counterfactual Prediction

Table A6: **Comparison with baseline methods on Causal-VidQA.** We re-train all methods using the same CLIP features using the officially released code. We also provide the results that use the glove features. '†' indicates the result re-implemented by the official code.

| Methods | QA | Features | Test_C | | | Avg |
|---|---|---|---|---|---|---|
| | | | Q → A | Q → R | Q → AR | |
| EVQA Antol et al. (2015) | | Glove | 27.72 | 27.57 | 10.63 | 21.97 |
| EVQA† Antol et al. (2015) | | CLIP | 28.05 | 28.05 | 10.09 | 22.00 |
| CoMem Gao et al. (2018) | | Glove | 42.97 | 42.24 | 22.25 | 35.82 |
| CoMem† Gao et al. (2018) | | CLIP | 45.70 | 47.60 | 25.44 | 39.83 |
| HME Fan et al. (2019) | | Glove | 35.29 | 34.19 | 15.34 | 28.27 |
| HME† Fan et al. (2019) | Yes | CLIP | 45.48 | 46.51 | 24.74 | 39.25 |
| HCRN Le et al. (2020) | | Glove | 43.69 | 43.47 | 22.75 | 36.64 |
| HCRN† Le et al. (2020) | | CLIP | 44.26 | 45.64 | 24.35 | 38.49 |
| HGA Jiang & Han (2020) | | Glove | 44.00 | 44.04 | 23.63 | 37.22 |
| HGA† Jiang & Han (2020) | | CLIP | 45.28 | 46.80 | 24.81 | 38.77 |
| B2A Park et al. (2021) | | Glove | 45.12 | 44.99 | 25.29 | 38.47 |
| B2A† Park et al. (2021) | | CLIP | 47.41 | 48.74 | 27.39 | 41.51 |
| CLIP Radford et al. (2021) | No | CLIP | 29.95 | 29.51 | 11.48 | 23.73 |
| **LLCP** | | CLIP | **39.07** | **38.46** | **19.03** | **32.13** |

question:"*What will happen if [person_5] has ingrown hairs or cuts*". Initially, we classify its question type and determine the pertinent question word. Following this, we identify the conditional clause by pinpointing the word if'. This conditional clause then serves as the keywords necessary for responding to counterfactual prediction questions. Delving further, we can pinpoint key objects that act as the subject within the clause.

Table A7: **Examples of parsing reasoning questions on SUTD-TrafficQA and Causal-VidQA datasets**.

| Reasoning Question Type | Open-choice Attribution |
|---|---|
| **Question** | What could possibly cause this accident? |
| **Rules** | Identify the "What could" questions and Attribution question type. Identify the effect is "the accident". It means that the keyword is the answer candidates. |
| **Keywords** | In the answer candidates |

| Reasoning Question Type | Open-choice Attribution |
|---|---|
| **Question** | Which might be the reason for this accident? |
| **Rules** | Identify the "Which might be" questions and Attribution question type. Identify the effect is "the accident". It means that the keyword is the answer candidates. |
| **Keywords** | In the answer candidates |

| Reasoning Question Type | Open-choice Counterfactual Prediction |
|---|---|
| **Question** | What would happen if [person_2] dropped the plates? |
| **Rules** | Identify the "What would happen" questions and Counterfactual question types. Parse the question and find the conditional clause introduced by "if". |
| **Keywords** | [person_2] dropped the plates |

| Reasoning Question Type | Open-choice Counterfactual Prediction |
|---|---|
| **Question** | What will happen if [person_5] has ingrown hairs or cuts? |
| **Rules** | Identify the "What will happen" questions and Counterfactual question types. Parse the question and find the conditional clause introduced by "if". |
| **Keywords** | [person_5] has ingrown hairs or cuts |

| Reasoning Question Type | Counterfactual True-or-false |
|---|---|
| **Question** | Will an accident happen if the vehicle in the front suddenly stop? |
| **Rules** | Identify the "Will ... happen if" questions and True-or-false question types. Parse the question and find the pattern "... happen if ...". Parse the conditional clause and find the noun "vehicle". |
| **Keywords** | Vehicle |

| Reasoning Question Type | Counterfactual True-or-false |
|---|---|
| **Question** | Would the accident still happen if all vehicles drive in their correct lane? |
| **Rules** | Identify the "Would ... happen" questions and True-or-false question types. Parse the question and find the pattern "... happen if ...". Parse the conditional clause and find the nouns "vehicles" and "lane". |
| **Keywords** | Vehicles and Lane |

| Reasoning Question Type | Introspection True-or-false |
|---|---|
| **Question** | Could the accident be prevented if all vehicles keep a safe distance away from one another? |
| **Rules** | Identify the "Could ... prevented if" questions and True-or-false question types. Parse the question and find the pattern "... prevented if ...". Parse the conditional clause and find the nouns "vehicles" and "distance". Align with the logic: cause → accident. Got "vehicles" and "distance" |
| **Keywords** | Vehicles and Distance |

