# OpenReview forum: "LLCP: Learning Latent Causal Processes for Reasoning-based Video Question Answer"
_ICLR.cc/2024/Conference — ICLR 2024 poster_

### Official Review · Reviewer_U5ds · 2023-10-26

**Soundness:** 3 good
**Presentation:** 3 good
**Contribution:** 4 excellent
**Rating:** 8
**Confidence:** 5

**Summary:**

This work introduces a weakly-supervised approach for reasoning about accidental events, leveraging causal representation learning combined with experimental tests on the derived causal variables. These causal variables, when integrated with CLIP, are aligned into natural language, forming an explainable AI (XAI) system. The proposed method undergoes rigorous analysis on simulated data and demonstrates comparable, performance to recent supervised techniques when tested on two real-world datasets.

**Strengths:**

1. The paper is well-structured, appealing to a broad readership.
2. It effectively applies causal learning to a real-world scenario, using a robust implementation with statistical tests on TMGM-derived causal variables.
3. The technique innovatively connects causal variables and natural language through CLIP, potentially reshaping standards in explainable AI (XAI).
4. Evaluation is thorough, including simulations and comparisons with supervised methods.
5. Impressively, the weakly-supervised approach matches the performance of supervised methods.

**Weaknesses:**

The authors primarily use a generative model for potential causes and treat language as a secondary filter. This approach however neglects language cues. Is that possible to have the lanuage involved for the genertive process? While the experiments show good improvements, could the author clarify the motivation of unsupervised generative approach over the supervised ones.

**Questions:**

See the weakness.

---

> ### Author Response · Authors · 2023-11-20
> **Response to Reviewer U5ds**
>
> Dear Reviewer U5ds, we would like to express our deep gratitude for the valuable feedback and helpful suggestions provided on our paper, as well as for the time you devoted to reviewing it. Below, we have addressed each of your comments and suggestions.
>
> > **W1:** The authors primarily use a generative model for potential causes and treat language as a secondary filter. This approach however neglects language cues. Is that possible to have the lanuage involved for the genertive process?
>
> **A1:** We thank you for the thoughtful and constructive reviews, which enhance the content depth of our paper. The language cues definitely can be involved in the generative models. However, it requires the language signal to be aligned with the video event, i.e., it requires annotations. Otherwise, the random textual will not benefit the training of the generative models.
> In this paper, we would like to explore a more challenging and realistic scenario, such as no textual annotations are used for model training. Below, we introduce the rationality and motivation of this scenario
>
>
> **Rationality**
> Intuitively, our human beings can effectively gain reasoning ability without cross-modality supervision by only leveraging the induction of observations.
> Theoretically, we can identify the latent variables and causal relations in an unsupervised way. Both of them serve as the support of the rationality of our task to explain how to learn causal relations without textual supervision.
>
> **Motivation**
> Here we show that real-world applications can benefit from this setting.
> First, on the difficulty of data collection, the annotated textual pairs are more than the unannotated videos. It would raise extra costs to annotate the data with neural language. Second, the model is easy to be misled by the short-cut in the textual annotations. For example, for all questions in the training set beginning with "What is the reason", the answer is similar, such as "White car speeding".  Then the model is easy to learn this shortcut and responds to the "White car speeding" for any questions beginning with "What is the reason". This issue caused by the training data is called language bias, which is widely discussed in the field of question answering[R1, R2].
>
> Please find the discussion in Appendix A3.3.
>
> [R1] Multifactor Sequential Disentanglement via Structured Koopman Autoencoders. Nimrod Berman and Ilan Naiman and Omri Azencot. The Eleventh International Conference on Learning Representations,2023.
>
> [R2] Klindt D, Schott L, Sharma Y, et al. Towards nonlinear disentanglement in natural data with temporal sparse coding[J]. arXiv preprint arXiv:2007.10930, 2020.
>
> **Experiments**
> In light of your suggestions, we further conducted an experiment that involved the annotated language signals in the training process of generative models, i.e., using our LLCP as a plug-in module to identify the latent variables and learn the causal relations.
>
> To learn the latent causal representation, we add two extra modules to the HCRN baseline, including an MLP Decoder and a Prior Aligner. The MLP Decoder uses HCRN's feature output to generate current observations, while the Prior Aligner assists in calculating KL divergence, essential for modeling transition functions and ensuring conditional independence with the prior. LLCRP, following HCRN's approach, first constructs latent causal representations using an encoder network identical to HCRN. Then, similar to HCRN, the output latent variables are learned with the supervision from language side. To plug the LLCP, we also apply a decoder to generate the observations from the latent space. To ensure the independence of latent variables, LLCRP imposes a KL divergence constraint between the posterior distribution of these variables and a predefined prior distribution. Finally, we also employ a reconstruction loss between the ground truth and generated observations.
>
> As shown in the Table below, we report the performance of LLCRP on SUTD-TrafficQA and compare it with the original HCRN method. We can observe that learning the causal representation and process helps improve the accuracy of reasoning-based VideoQA tasks. On average, LLCRP achieves an improvement of almost 5% over the baseline HCRN.
>
> |     | B     | A     | I     | C     | F     | R     | Avg   |
> |-----|-------|-------|-------|-------|-------|-------|-------|
> | HCRN             | 34.17 | 50.29 | 33.4  | 40.73 | 44.58 | 50.09 | 36.26 |
> | LLCRP             | 38.95 | 44.98 | 32.43 | 48.64 | 41.7  | 47.16 | 41.22 |
>
>
> > **W2:** While the experiments show good improvements, could the author clarify the motivation for the unsupervised generative approach over the supervised ones.
>
> **A2:** We highly appreciate the suggestion, which helped us clarify the motivation of our paper. As mentioned above, the motivation of an unsupervised generative approach lies in the lower annotation cost and more robustness for language bias. Please refer to A1 for details.

---

> ### Author Response · Authors · 2023-11-21
> **Have the concerns been adequately addressed in the response and revision?**
>
> Dear Reviewer U5ds,
>
> Thank you for dedicating time to review and provide feedback on our submission. We hope our response and revised work effectively address your concerns. If there are additional matters you'd like us to consider, we eagerly await the opportunity to respond.
>
> Best regards,
>
> Authors of submission 910

---

### Official Review · Reviewer_DRzn · 2023-10-31

**Soundness:** 2 fair
**Presentation:** 3 good
**Contribution:** 2 fair
**Rating:** 3
**Confidence:** 4

**Summary:**

This paper introduces a video reasoning framework called Learning Latent Causal Processes (LLCP). LLCP first makes use of a conditional VAE framework to extract the spatio-temporal dynamics of objects in videos in a self-supervised manner. These extracted dynamics then serve as weak supervision to analyze the relationships between events during inference time, facilitating the identification of root cause of accidents and counterfactual effects. Tested on both synthetic and real-world benchmarks, the proposed method demonstrates advantage over baselines while still underperforming supervised methods.

**Strengths:**

- It is well-motivated and a desired but missing capability in AI systems to model causal relationships between objects/events in video understanding.

- The implementation seems straightforward.

- The proposed method demonstrates promising results when compared with some baseline methods and its supervised counterparts on both synthetic and real-world benchmarks.

**Weaknesses:**

- The novelty of the paper is limited. While the paper draws motivation from causal modeling, both the theoretical analysis and the implementation of TMGM using conditional VAEs do not adequately realize the causal processes in Eq. 1.

- It is not clear how the authors enforce the independence constraints, both spatially and temporally, on the noise variables by the three subnetworks f_N, f_H and f_E. I would say the fusion of these three networks encodes spatio-temporal dynamics across objects (agents) but it is unclear if this has any bearing on causal relations.

- Experiments only compare LLCP with weaker variants of VAEs which offer little to no  temporal modeling. In addition, the proposed method largely underperforms supervised methods as indicated by Table 5 and Table 6.

**Questions:**

Apart from my concerns in the Weakness section. I have some other questions:

- Regarding the first task of root cause analysis, is there any reference for the definition of root causes in Sec. 3.2? The provided definition seems to be applicable with only anomaly/outlier detection.

- In Sec. 3.4, what happens if we formulate counterfactual questions in an open-ended format, i.e. “what if”?

- The outlier analysis at inference times relies on extracted features from pretrained models (i.e., CLIP) which can be brittle if tested data is outside of the trained distribution. In addition, using language parser to parse user queries is not ideal as well. Have you thought of an alternative solution towards this?

---

> ### Author Response · Authors · 2023-11-20
> **Response to Reviewer DRzn Part 1**
>
> Dear Reviewer DRzn, we highly appreciate your time dedicated to reviewing this paper. Following your comments and suggestions, we have further highlighted the contributions, clarified the model details, extended the experimental results, and provided more justifications. We modified the manuscript and appendix accordingly. Please find our point-to-point response below. Hope it can address all your concerns.
>
> > **W1:** The novelty of the paper is limited. While the paper draws motivation from causal modeling, both the theoretical analysis and the implementation of TMGM using conditional VAEs do not adequately realize the causal processes in Eq. 1.
>
> **A1:** We appreciate this comment which helped us clarify both theoretical and technical details and highlight our contributions.  We would like to highlight that under the assumptions we make, our TMGM model is able to learn the causal processes. We have included detailed proof in the revised Appendix A2.
> We have conducted an experiment to verify whether we can learn the causal processes in Appendix A6.3.
> We also analyzed the relations between motivation and implementation in the following. The discussion on how the implementation details can constrain/leverage these assumptions
>
> 1. **Key Motivation**
>    Our key motivation is to learn the model by answering reasoning questions without the usage of textual annotations. To achieve that, our LLCP consists of two stages, one is learning the causal processes with the temporal multivariate generative model, and the other is applying the discovered causal processes to conduct inference for answering reasoning-based questions. The first stage relies on the assumptions of Granger Causality, and the latter one leverages the shared feature spaces and causal inference.
>
> 2. **Granger Causality in Learning the Causal Processes**
>    - **Assumption:** We assume that the past values of one variable can predict the future values of another, indicating a directional relationship in a time-series context and implying no instantaneous effects in the temporal dynamics. Variables $\mathbf{z}_t$ are independent given the historical state $Pa(\mathbf{z}_t)$, as aligned with our data generation process.
>    - **Implementation:** To substantiate this, we employ auto-regression based on the concept of Granger Causality. This approach predicates future events on historical data, validating our assumption about predictability. The **identification of our proposed model** is further supported by a proposition and proof in Appendix A2.
>
> 3. **Causal Inference**
>    - **Root cause analysis** We conduct the root cause analysis by identifying the changed local causal mechanisms.  The insight is if one variable's change cannot be explained by its parents, then it is the root cause of the change.
>    - **Counterfactual** Counterfactual inference is an inference method that leverages the causal structures to predict the possible outcome in the counterfactual conditions.
>
> **W2:** It is not clear how the authors enforce the independence constraints, both spatially and temporally, on the noise variables by the three subnetworks f_N, f_H and f_E. I would say the fusion of these three networks encodes spatio-temporal dynamics across objects (agents) but it is unclear if this has any bearing on causal relations.
>
> **A2:** Thanks very much for this question. It definitely helped improve the readability of our paper. We respectively answer the questions in terms of independence and model-fusion below. We have revised the manuscript accordingly to make the details clearer as shown in Section 2.1.
>
> **Independence.** We enforce the independence by the KL divergence loss in Equation (3) in the revised version (original Equation (2) ).
> Specifically, we constrain the learned posterior distribution with a prior distribution where $  p(\bf{\hat z}_t) | Pa(\bf{\hat z}_t) )= \prod _{i=1}^{M} p({\bf{\hat z}_t}_i) | Pa({\bf{\hat z}_t}_i) ) $ which implies the following conditional independent properties. Given the historical information, the variables in the current time step are independent. By this, with the log function, we have
>  $  \log(p(\bf{\hat z}_t) | Pa(\bf{\hat z}_t) )) = \sum _{i=1}^{M} \log(p({\bf{\hat z}_t}_i) | Pa({\bf{\hat z}_t}_i) )) $.
> With the KL loss, we encourage the posterior also with these conditional independent properties.
>
> **Subnetworks.** By the design of three subnetworks, we decompose the causal processes into local mechanisms. First, this indicates prior knowledge that the transition functions are different for historical states, neighborhood interactions, and environmental clues. By this decomposition, we respectively learn each submodule leveraging the modularity property of the causal mechanisms, which makes the learning process easier.

---

> ### Author Response · Authors · 2023-11-20
> **Response to Reviewer DRzn Part 2**
>
> **W3:** Experiments only compare LLCP with weaker variants of VAEs which offer little to no temporal modeling. In addition, the proposed method largely underperforms supervised methods as indicated by Table 5 and Table 6.
>
> **A3** We are grateful for your careful review and constructive suggestions to improve the completeness of our experiments. In light of your suggestion, we have further considered several latest VAE variants like SKD [R1] and SlowVAE [R2], which are shown in the following table. According to the experiment results, we can find that our method can still enjoy significant improvement. Please refer to Table A3 for details.
>
> | Models   | function change |          | Structure change |          |
> |----------|-----------------|----------|------------------|----------|
> |          | recall          | f1 score | recall           | f1 score |
> | LSTM VAE | 52.11           | 27.81    | 37.75            | 20.78    |
> | VAE      | 51.92           | 27.33    | 31.08            | 19.35    |
> | SlowVAE  | 54.41           | 29.03    | 38.12            | 21.39    |
> | SKD      | 58.93           | 34.76    | 35.72            | 19.79    |
> | LLCP     | 63.34           | 63.01    | 45.04            | 44.19    |
>
> For the comparison with supervised learning methods, we would like to respectfully highlight that our method doesn't use any textual annotations, which causes the results to appear lower. For a more fair comparison, we also compared LLCP with other unsupervised learning such as CLIP [R3] and OpenFlamingo [R4] and found a significant improvement. Please refer to Appendix A7.2 for more details.
>
> | Model                       | Accuracy |
> |-----|-------------|
> | OpenFlamingo-3B (2 examples)   | 28.7        |
> | OpenFlamingo-3B (4 examples)   | 29.1        |
> | VidIL                       | in processing  |
> | CLIP                        | 27.7        |
> | LLCP                        | 33.7        |
>
> In addition, when further utilizing the textual annotations, we outperformed other supervised learning methods, such as the baseline HCRN, by a large margin (5\%). Please kindly refer to Appendix A5 and Table A2 for the implementation details and experimental results.
>
> |                    | B     | A     | I     | C     | F     | R     | Avg   |
> |--------------------|-------|-------|-------|-------|-------|-------|-------|
> | HCRN               | 34.17 | 50.29 | 33.40  | 40.73 | 44.58 | 50.09 | 36.26 |
> | LLCRP              | 38.95 | 44.98 | 32.43 | 48.64 | 41.70  | 47.16 | 41.22 |
>
>
>
> [R1] Multifactor Sequential Disentanglement via Structured Koopman Autoencoders. Nimrod Berman and Ilan Naiman and Omri Azencot. The Eleventh International Conference on Learning Representations,2023.
>
> [R2] Klindt D, Schott L, Sharma Y, et al. Towards nonlinear disentanglement in natural data with temporal sparse coding[J]. arXiv preprint arXiv:2007.10930, 2020.
>
>
> >**W4(Q1):** Regarding the first task of root cause analysis, is there any reference for the definition of root causes in Sec. 3.2? The provided definition seems to be applicable with only anomaly/outlier detection
>
> **A4** Thanks for your question which helped improve the readability.
> The "root cause" in root cause analysis is defined as the fundamental, underlying, system-related reason to explain the failure. It denotes that changing this factor can solve the problem from the root, which implies the changing of this factor cannot be explained by its cause variables[R3, R4].
>  In this paper, we consider the "root cause" from the perspective of causality and apply this to answer cause-based reasoning questions.
>
> [R3]Tariq S Abdelhamid and John G Everett. Identifying root causes of construction accidents. Journal of construction engineering and management, 126(1):52–60, 2000
>
> [R4] OSHA Fact Sheet. The importance of root cause analysis during incident investigation. URL https://www.osha.gov/Publications/OSHA3895.pdf, 2016.

---

> ### Author Response · Authors · 2023-11-20
> **Response to Reviewer DRzn Part 3**
>
> >**W5(Q2):** In Sec. 3.4, what happens if we formulate counterfactual questions in an open-ended format, i.e. “what if”?
>
> **A5** Thanks a lot. Similar to conventional VideoQA methods, the differences between the multi-choice and open-ended formats lie in the predictor module. We use classifier head and generative head respectively for multi-choice and open-ended formats. It indicates that we can train or pre-train a generative head to allow open-ended questions.
>
>
> > **W6(Q3):** The outlier analysis at inference times relies on extracted features from pretrained models (i.e., CLIP) which can be brittle if tested data is outside of the trained distribution. In addition, using language parser to parse user queries is not ideal as well. Have you thought of an alternative solution towards this?
>
> **A6** We appreciate this question which improved the potential of our method for real-world applications. We have two potential strategies to solve the possible OOD data. First, the CLIP model shows the excellent transferability [R5]. We can efficiently conduct the prompt learning or adapter tuning to align the pretrained CLIP into the test data distribution. Second, we may follow the conventional supervised methods to learn cross-modality matching and take the LLCP as the plugging module to learn causal presentations.  As shown in Appendix A5, we find that the LLCP can benefit from the annotations to achieve better performance, which shows the lights to solve the difficulty caused by no annotations.
>
> [R5] Zhou K, Yang J, Loy C C, et al. Learning to prompt for vision-language models[J]. International Journal of Computer Vision, 2022, 130(9): 2337-2348.

---

> ### Author Response · Authors · 2023-11-21
>
> Dear Reviewer DRzn
>
> We sincerely appreciate the time and effort you dedicated to reviewing our submission and providing insightful comments. We eagerly await your assessment of our response and revisions, ensuring they effectively address your concerns. Should you have any additional points of consideration, we are eager to address them and welcome the opportunity for further dialogue.
>
> Thank you for your valuable feedback.
>
> Best regards,
>
> Authors of submission 910

---

> ### Author Response · Authors · 2023-11-22
> **Could you please let us know whether our responses and updated submission properly addressed your concern?**
>
> Dear Reviewer DRzn,
>
> We express our sincere gratitude for taking the time to review our manuscript. Your suggestions regarding the the justifications and discussions of our article have greatly contributed to improving its quality. We have made detailed revisions to the manuscript and addressed your questions in our response. We hope that our answers have addressed any concerns you had regarding our work. The end of the rebuttal is coming. Your feedback is vital to us, and any response would be further appreciated.
>
> Best Regards
>
> Authors of submission 910

---

> ### Author Response · Authors · 2023-11-23
> **Could you please kindly let us know whether our responses and revisions properly addressed your concerns**
>
> Dear Reviewer DRzn
>
> We understand you are busy and appreciate your time. Here we are re-sending a previous message to make sure you see it.  To ensure our response has fully addressed your concerns, we would greatly appreciate your feedback on the recent revisions and clarifications we provided, including the added experiments. If there are any remaining concerns or questions, we are eager to address them promptly.
>
> Thanks a lot for all your effort you made in this paper.
>
> Best regards,
>
> Authors of Submission 910

---

### Official Review · Reviewer_naVu · 2023-10-31

**Soundness:** 2 fair
**Presentation:** 2 fair
**Contribution:** 3 good
**Rating:** 6
**Confidence:** 3

**Summary:**

This paper focused on the task of reasoning-based video QA. The challenge is the lack of causal relations in the cross-modal matching pipelines. This work employs a temporal multivariate generative model to understand the latent causal patterns, which can be used to answer caused-based and effect-based questions about videos. The experiments are conducted on two simulation datasets and two real-world VideoQA datasets.

**Strengths:**

+ The proposed model can achieve comparable performance to supervised methods while no paired textual annotations are used. The proposed framework can answer both cause-based (i.e., accident attribution) and effect-based (i.e., counterfactual prediction) reasoning questions. It shows the potential of self supervision.

+ The motivation of explore latent causal relations for video QA makes sense and is interesting.

**Weaknesses:**

- The presentation is not satisfying enough. It is hard for me to figure out the connection between motivation and implementation details, especially how the method can guarantee that the learned relation is causal rather than temporal, and the terminologies of variables. Please see "questions" for detailed comments.

- Another main concern is whether the model is causality-based or just capture temporal relations. In addition to the concept of causal relation, I didn't see how the implementation reflect the tools of causal inference or causal reasoning. Therefore, I am wondering whether the causal understanding ability is over-claimed. Experimental results can verify the ability empirically, but theoretical explanations or guarantees are missing.

- The ablation studies in Table 4 mainly demonstrate the contributions of sub-networks rather than the role of historical states and environment. What if we replace the historical states and environment with wrong ones? What is the performance then? That would show that the performance drop is due to the lack of visual information rather than fewer parameters.


---

After rebuttal: the authors' responses addressed my concerns including causal theory, ablation studies, and clarification on terminologies. I didn't carefully check the details of Granger Causality as it is out of my knowledge but assume that the usage of Granger Causality is correct.

**Questions:**

1. What does the red node in Figure 1 (b) mean? What it is red during test pipeline but not highlighted during training pipeline?

2. The abstract mentioned that the proposed LLCP employs a temporal multivariate generative model to understand the causal patterns. I am not aware of how the *temporal* model can discover *causal* relations. I am wondering how to guarantee that the learned pattern are causal relations rather than temporal correlations using the so-called temporal model? In the method part, I am not aware of how the learned pattern are causality rather than correlation.

3. I didn't find strict definitions of historical state, neighborhoods, and environment variables. Could the authors provide a precise and accurate definition, or give examples of these three variables? For example, are they features of a single frame, an object, or a set of tracked frames? How does the object tracking model obtain these three variables in temporal and spatial dimensions? It seems that Figure 4 provide an example, but it appears a bit late when I was reading Sec. 3.1 but didn't find the examples.

4. In Ex. (2), is x_{t,i} an image (or region of interest) or feature vector?

5. According to Figure 4, the question seems to be the input of text side rather than video side. In this case, how can we determine the object of interest when extract the visual information and make the prediction of visual states? Is it reasonable to extract question-independent visual feature to answer the question?

6. What does the arrows mean in Figure 5? Are they drew manually or automatically estimated by the model?

---

> ### Author Response · Authors · 2023-11-20
> **Response to Reviewer naVu, Part 1**
>
> Dear Reviewer naVu, we sincerely appreciate your suggestions which have helped improve our readability. We have included more justifications and discussions in the revised version, and extended the experiments of ablation studies. Please refer to the revised paper and appendix for details.
>
> > **W1:** The presentation is not satisfying enough. It is hard for me to figure out the connection between motivation and implementation details, especially how the method can guarantee that the learned relation is causal rather than temporal, and the terminologies of variables.
>
> **A1:** Thanks for your valuable comments which have helped clarify our motivation, assumptions, and technical details.
> In light of your suggestions, we list the assumptions and discuss how the implementation details can constrain/leverage these assumptions in Appendix A3.1.
> Besides, we further provide the theoretical guarantee of why the learned model follows the Granger Causality in Section 2.1 and Appendix A2.
>
> Our method's connection between motivation and implementation is deeply rooted in the core assumptions of the **LLCP** model, which are crucial for understanding causality in the temporal dynamic system.
> 1. **Key Motivation**
>    Our key motivation is to learn the model for answering reasoning questions without the usage of textual annotations. To achieve that, our LLCP consists of two stages, one is learning the causal processes with the temporal multivariate generative model, and the other is applying the discovered causal processes to conduct inference for answering reasoning-based questions. The first stage relies on the assumptions of Granger Causality, and the latter one leverages the shared feature spaces and causal inference.
>
> 2. **Granger Causality Principle**
>    - **Assumption:** We assume that the past values of one variable can predict the future values of another, indicating a directional relationship in a time-series context and implying no instantaneous effects in the temporal dynamics. Variables $\mathbf{z}_t$ are independent given the historical state $Pa(\mathbf{z}_t)$, as aligned with our data generation process.
>    - **Implementation:** To substantiate this, we employ auto-regression based on the concept of Granger Causality. This approach predicates future events on historical data, validating our assumption about predictability. The identification of our proposed model is further supported by a proposition and proof in Section A2.
>
> 3. **Shared Feature Spaces in Visual and Textual Domains:**
>    - **Assumption:** We posit that the feature spaces for visual and textual domains are shared, facilitating the matching of the same variables across these domains.
>    - **Implementation:** To corroborate this, we integrate the pre-trained CLIP model as a feature extractor for both video objects and textual elements (questions and answers). Its application ensures that both visual and textual data are represented in congruent feature spaces, thus affirming the feasibility of matching variables across these domains.
>
> 4. **Causal Inference**
>    - **Root cause analysis** We conduct the root cause analysis by identifying the changed local causal mechanisms.  The insight is if one variable's change cannot be explained by its parents, then it is the root cause of the change.
>    - **Counterfactual** Counterfactual inference is an inference method that leverages the causal structures to predict the possible outcome in the counterfactual conditions.
>
> **Terminologies of variables**
> We guess that the reviewer is curious about the terminologies of variables. This term denotes an object with randomness (probability distribution). Here, our observed variables are the features of objects, such as vehicles, whose randomness comes from different characteristics and states. While the latent variables are hidden, which contribute to the data generation process.

---

> ### Author Response · Authors · 2023-11-20
> **Response to Reviewer naVu, Part 2**
>
> > **W2:** Another main concern is whether the model is causality-based or just capture temporal relations. In addition to the concept of causal relation, I didn't see how the implementation reflect the tools of causal inference or causal reasoning. Therefore, I am wondering whether the causal understanding ability is over-claimed. Experimental results can verify the ability empirically, but theoretical explanations or guarantees are missing.
>
>
> **A2:** We are grateful for the suggestions which help clarify our theoretical soundness.
> We would like to respectfully highlight that the proposed LLCP framework is a causal framework, which consists of two stages: causal discovery (learn the causal structures) and causal inferences (use the causal structures).
> In light of your suggestion, we have included the identification theorem to show that our proposed model can learn the causal relations by proposition and proof in Appendix A2.
>
> **Learn the causal structures**
> We assume that data follows the Granger Causality, which implies that the past values of one variable can be used to predict the future values of another, suggesting a directional relationship between these variables in a time series context. With this assumption, we provide proof to show that the learned causal relations are identifiable. Please refer to Appendix A2 and the revised Section 2.1 for details.
>
> **Leverage the causal structures**
> In the stage of causal inference, we propose to conduct the root cause analysis and counterfactual predictions respectively for cause-based and effect-based reasoning tasks. The former leverages the modularity of the causal mechanisms to locate the root cause by matching the observed agents with the local generative process. The counterfactual prediction is also a classical causal inference tool to provide personalized prediction under counterfactual conditions. Please kindly refer to Chapter 7, Section 1 of Judea Pearl's book “Causality” [R1], for more details.
>
> [R1] Pearl J. Causality[M]. Cambridge university press, 2009.
>
>
> > **W3:** The ablation studies in Table 4 mainly demonstrate the contributions of sub-networks rather than the role of historical states and environment. What if we replace the historical states and environment with wrong ones? What is the performance then? That would show that the performance drop is due to the lack of visual information rather than fewer parameters.
>
>
> **A3:** We greatly appreciate this suggestion which improved the completeness of our ablation studies. In light of your suggestions, we further conducted the experiments by replacing the historical states and environment with random vectors.
> As shown in Table 3 in the revised version, we summarized the performance of all versions of LLCP, including using random features of historical states and environments.
> First, using random historical feature $f_{N}$ and environmental feature $f_{E}$ hurts the performance, which demonstrates both these two modules help learn the causal generation process.
> Second, we found that the parameters also influence the performance since the performance of using the random feature is a bit better than the total removal.
> Thanks again for your insightful advice!
>
> > **W4(Q1):** What does the red node in Figure 1 (b) mean? What it is red during test pipeline but not highlighted during training pipeline?
>
> **A4:** Thank you very much for this question. It helps us to further clarify the illustration and improve the readability. The red node in Figure 1 (b) means the root cause (the agent causing the accident). We have added this in the captions of the revised version. For the training stage, we learn the causal mechanisms from regular videos in which **no** accident is there. In the inference stage, we apply the causal inference to identify the root cause and answer the corresponding questions.
>
> > **W5(Q2):** The abstract mentioned that the proposed LLCP employs a temporal multivariate generative model to understand the causal patterns. I am not aware of how the temporal model can discover causal relations. I am wondering how to guarantee that the learned pattern are causal relations rather than temporal correlations using the so-called temporal model? In the method part, I am not aware of how the learned pattern are causality rather than correlation.
>
> **A5:** Thanks for your questions.  Please refer to the Answer 2 above for the details. We assume the variables in temporal dynamics follow the Granger Causality, which implies that the past values of one variable can be used to predict the future values of another, suggesting a directional relationship between these variables in a time series context. Thus, we can directly obtain the causality (directional) rather than correlation (unidirectional). We provided a theoretical guarantee in Appendix A2 and Section 2.1 in the revised version.

---

> ### Author Response · Authors · 2023-11-20
> **Response to Reviewer naVu, Part 3**
>
> > **W6(Q3):** I didn't find strict definitions of historical state, neighborhoods, and environment variables. Could the authors provide a precise and accurate definition, or give examples of these three variables? For example, are they features of a single frame, an object, or a set of tracked frames? How does the object tracking model obtain these three variables in temporal and spatial dimensions? It seems that Figure 4 provide an example, but it appears a bit late when I was reading Sec. 3.1 but didn't find the examples.
>
> **A6:** Thanks a lot for the valuable suggestions which improved the readability of our paper. Here, all of the historical state, neighborhoods, and environment variables are the variables in the temporal dynamic system.  The variables here can be understood as the agents in traffic events, such as vehicles, pedestrians, or traffic signals.
> Below are the detailed definitions:
> For a given variable $x_{t, i}$, its historical state is the variable with the same identity but in the past frame, i.e., $x_{t', i} | t'<t$. Similarly, its neighborhoods denote other variables (not experimental variables) in the previous frame, such as $x_{t',j}| t'<t, j \neq i$. The environment variables denote the environmental context, such as traffic lights, zebra crossing, and traffic signals. We have included these detailed definitions in the revised Section 2.1 (Section 3.1 in the revised paper).
>
> > **W7(Q4):** In Ex. (2), is x_{t,i} an image (or region of interest) or feature vector?
>
> **A7:** Thanks for this question.  $x_{t,i}$ is a feature vector, which is obtained by the ROI pooling on the CLIP feature tensor for the detected bounding box of a certain object, such as a vehicle.
>
>
> > **W8(Q6):** What does the arrows mean in Figure 5? Are they drew manually or automatically estimated by the model?
>
> **A8:** The arrows denote the learned causal relations. $A \rightarrow B$ denotes that A causes B.  We manually draw the lines on the picture according to the learned causal structure.

---

> ### Author Response · Authors · 2023-11-20
> **Response to Reviewer naVu, Part 4**
>
> > **W9(Q5):** According to Figure 4, the question seems to be the input of text side rather than video side. In this case, how can we determine the object of interest when extract the visual information and make the prediction of visual states? Is it reasonable to extract question-independent visual feature to answer the question?
>
> **A9:** We sincerely appreciate this question, which led to a new light to further improve our methods.  In light of your suggestion, we included a justification for the rationality and motivation for training without textual annotations in Appendix A3.3.
> Please note that the usages of textual information in training and inference are different. We respectfully believe the inference needs the questions to identify the focus of interest, while the learning process can benefit from annotations but not necessary. We also extended our method to leverage the QA annotations. The detailed method, implementations, and experiments can be found in Appendix A5.
>
>
> **Training Without Textual Pairs**
> It would be really good if we could always have enough annotated question-answering pairs as the supervision. It would help the learning of the reasoning model. However. the annotated question-answering pairs are not easy to obtain, especially for tasks requiring domain knowledge such as traffic accident attribution.
> For humans, learning to reason can be effectively achieved without cross-modality supervision, relying solely on observation induction within shared feature spaces. This concept aligns with causality theories[R2 R3], suggesting the feasibility of uncovering latent variables and causal relationships unsupervised.
> Additionally, models trained with textual annotations are susceptible to language biases, where they might learn shortcuts based on linguistic patterns rather than understanding the content. For instance, if all questions in the training set beginning with "What is the reason" have similar answers, the model might generalize inaccurately, associating any question with this format to a specific answer, such as "White car speeding", regardless of context. This phenomenon, known as language bias, has been extensively discussed in question-answering research[R4, R5].
>
> **Use of Text Queries in the Inference Phase**
> Despite the avoidance of textual pairs in training, text queries play a crucial role during the inference phase. Our model, trained exclusively on visual data, can identify the visual causal relations of variables. During inference, textual queries are employed to first identify the local causal mechanisms we are interested in. Then, we leverage the causal relations to answer the detailed question. For example, if two accidents occur in the same video, our model will find causal mechanisms with two root causes. Then the query questions can help to identify the root cause we are more interested in.
>
> **Extend the Method to Leverage QA Pairs**
> To learn the latent causal representation, we add two extra modules to the HCRN baseline, including an MLP Decoder and a Prior Aligner. The MLP Decoder uses HCRN's feature output to generate current observations, while the Prior Aligner assists in calculating KL divergence, essential for modeling transition functions and ensuring conditional independence with the prior. LLCRP, following HCRN's approach, first constructs latent causal representations using an encoder network identical to HCRN. Then, similar to HCRN, the output latent variables are learned with the supervision from language side. To plug the LLCP, we also apply a decoder to generate the observations from the latent space. To ensure the independence of latent variables, LLCRP imposes a KL divergence constraint between the posterior distribution of these variables and a predefined prior distribution. Finally, we also employ a reconstruction loss between the ground truth and generated observations.
>
> As shown in the Table below, we report the performance of LLCRP on SUTD-TrafficQA and compare it with the original HCRN method. We can observe that learning the causal representation and process helps improve the accuracy of reasoning-based VideoQA tasks. On average, LLCRP achieves an improvement of almost 5% over the baseline HCRN.
>
> | |B|A|I|C|F|R|Avg|
> |-|-|-|-|-|-|-|-|
> |HCRN|34.17|50.29|33.40|40.73|44.58|50.09|36.26|
> |LLCRP|38.95|44.98|32.43|48.64|41.70|47.16|41.22|
>
> [R2]Yao W, Sun Y, Ho A, et al. Learning temporally causal latent processes from general temporal data[C]. ICLR, 2021.
>
> [R3]Yao W, Chen G, Zhang K. Temporally disentangled representation learning[J]. NeurIPS, 2022, 35: 26492-26503.
>
> [R4] Desen Yuan. Language bias in visual question answering: A survey and taxonomy. arXiv preprint arXiv:2111.08531, 2021
>
> [R5] Zhiquan Wen, Guanghui Xu, Mingkui Tan, Qingyao Wu, and Qi Wu. Debiased visual question answering from feature and sample perspectives. NeurIPS, 34:3784–3796, 2021

---

> ### Author Response · Authors · 2023-11-21
> **Have the response and revision adequately address to your concerns?**
>
> Dear Reviewer naVu,
>
> Thanks for your time dedicated to carefully reviewing this paper. It would be further highly appreciated if you let us know whether our response and the change in the paper properly address your concerns, despite your busy schedule. Thanks a lot!
>
> With best regards,
>
> Authors of submission 910

---

> ### Author Response · Authors · 2023-11-22
> **Could you please kibndly let us know whether our responses and updated submission properly addressed your concerns?**
>
> Dear Reviewer naVu,
>
> We highly appreciate your time for reviewing this paper and all your efforts which have improved our readability. Follow your suggestions, we have tried our best to add more justifications and discussions in the revised version, and conduct the experiments. Due to the limited rebuttal discussion period, we hope any opportunity to response your possible concerns.  Please kindly let us know if you have any.
>
> Thanks again for all your efforts on this paper.
>
> Best regards,
> Authors of submission 910

---

> > ### Comment · Reviewer_naVu · 2023-11-22
> >
> > Thank the authors for the detailed responses, which addressed my concerns including causal theory, ablation studies, and clarification on terminologies. I didn't carefully check the details of Granger Causality as it is out of my knowledge but assume that the usage of Granger Causality is correct. I am happy to increase my rating to 6 and look forward to other reviewers' feedback.

---

> > > ### Author Response · Authors · 2023-11-22
> > > **Thanks for your feedback!**
> > >
> > > Dear Reviewer naVu,
> > >
> > > We would like to express our gratitude for your constructive feedback and approval of our work. Your insights into causal theory, ablation studies, and the need for clearer terminologies have been invaluable. We believe that incorporating your suggestions has significantly enhanced the quality of our submission. Thank you for your guidance and support throughout this process.
> > >
> > > Best regards,
> > >
> > > Authors of Submission 910

---

### Official Review · Reviewer_dMTZ · 2023-11-01

**Soundness:** 2 fair
**Presentation:** 3 good
**Contribution:** 2 fair
**Rating:** 5
**Confidence:** 4

**Summary:**

The Learning Latent Causal Processes (LLCP) framework introduces a novel approach to Video Question Answering (VideoQA) by focusing on causal reasoning rather than traditional cross-modality matching. LLCP utilizes a multivariate generative model to analyze spatial-temporal dynamics and trains through self-supervised local auto-regression, thus eliminating the need for annotated question-answer pairs. It adeptly handles accident attribution and counterfactual prediction tasks, identifying root causes and forecasting potential outcomes through modifications in variable embeddings.

**Strengths:**

1. The idea of learning causality from video is interesting.
2. The paper is overall presented clearly.
3, The authors made efforts on providing fair comparisons with existing methods.

**Weaknesses:**

1. The main concern is that the current setting is too far from realistic settings. The current evaluation setting is really more like hacking parts of existing datasets. The reviewer encourage the authors to make this work more complete.

a. It is too constraint to only evaluate the proposed method when it has no access to QA labels. If the obtain model really captures the casual relationship in videos, plugging it into existing methods in the supervised training setting can show a much more broad application of the proposed method.

b. It is also not realistic to exclude the text query from the training process since the fusion between visual and textual input is also the crucial design to really solve the VQA problem. For example, if there are two accidents going on in the video, the current framework will have systematical flaw as the model is not conditioned on the question text which specifies which accident it is about.

c. Despite the effort of re-training many existing methods, it is not well-justified why it is necessary to discard important features like motion or object as used in Causal-Vid-QA, which brings all the models to a low-performance scheme.

d. There is no proper comparison with methods that do not require QA data like but not including to [a,b,c]. The authors should also acknowledge and at least provide comparison with some of these relevant methods to really provide the audience a correct and comprehensive understanding of the relevant solution to this setting.

e. Once a and b are done, it is also necessary to provide additional comparison on broader VideoQA datasets to understand the importance of causal learning process in videos for broader videoQA tasks, which is really beneficial for the community.



[a] 🦩Flamingo: a Visual Language Model for Few-Shot Learning
[b] Language Models with Image Descriptors are Strong Few-Shot Video-Language Learners
[c] Language Models are Causal Knowledge Extractors for Zero-shot Video Question Answering


Minor:
1. Title, related work: Video Question Answer -> Video Question Answering.

**Questions:**

Please check weakness for details.

---

> ### Author Response · Authors · 2023-11-20
> **Response to Reviewer dMTZ, Part 1**
>
> Dear Reviewer dMTZ, we sincerely appreciate your informative feedback and helpful suggestions that helped enhance the integrity of our evaluations. We have added the new experiments following your suggestions, provided more discussions, and modified the paper and appendix accordingly. Please see our point-to-point response below.
>
>
> > **W1:** The main concern is that the current setting is too far from realistic settings. The current evaluation setting is really more like hacking parts of existing datasets. The reviewer encouraged the authors to make this work more complete.
>
> **A1:** We appreciate the suggestion which helped make this paper more complete and closer to realistic applications.
>
> **Make more complete experiments**
> In light of this suggestion and the following detailed points, we have included more comprehensive evaluations, and provided more discussions as well as justifications for the experimental designs. Specifically,
> 1) we extended our method by using QA annotations (as a plug-in module) and compared it with the baseline in Appendix A5;
> 2) we added the comparisons and discussions with the zero-shot multi-modal LLM models, such as OpenFlamingo, in Appendix A7.2;
> 3) we included more justifications on rationality and motivation for training without textual annotations in Appendix A3.3;
> 4) we clarified the experimental settings of Causal-VideoQA;
> 5) we analyzed a principle way to use LLCP for broader settings.
>
> **Justify the reality of the current setting**
> We appreciate your variable questions about the experimental settings and enjoy further discussions on its reality.
> We respectfully believe that our experimental settings focus on a more difficult and realistic scenario instead of a hacking of the existing datasets.
> Though textual annotations can be meticulously prepared for the training of the reasoning models to enhance the model's ability, in many applications, such as traffic accident reasoning, it is unrealistic to prepare large-scale well-organized textual annotations for the reasoning questions, since it is usually time-consuming, labor-intensive and requires expert knowledge.
> Therefore, it is realistic and important to address the reasoning QA problem in an unsupervised manner. However, there are no published datasets for this problem. To solve this problem, we employ the existing reasoning QA datasets with textual annotation and compare them with the existing supervised methods, which also shows the advantages of our methods.
>
> > **W2:** It is too constraint to only evaluate the proposed method when it has no access to QA labels. If the obtain model really captures the casual relationship in videos, plugging it into existing methods in the supervised training setting can show a much more broad application of the proposed method.
>
> **A2:** We are grateful for this suggestion which makes our evaluations more complete. In light of your suggestions, we proposed to plug LLCP into existing methods in the supervised training setting, such as the HCRN [1] method, and provided the experimental results on the SUTD-TrafficQA datasets. The proposed method plugging the LLCP learns the both latent causal representation and process, which is called LLCRP. Please kindly refer to Section A5 in the revised appendix for the detailed model design and experimental evaluations. Here, we provide a brief summary of LLCRP and the experimental comparison.
>
> **Implementation Details**
> To learn the latent causal representation, we add two extra modules to the HCRN baseline, including an MLP Decoder and a Prior Aligner. The MLP Decoder uses HCRN's feature output to generate current observations, while the Prior Aligner assists in calculating KL divergence, essential for modeling transition functions and ensuring conditional independence with the prior. LLCRP, following HCRN's approach, first constructs latent causal representations using an encoder network identical to HCRN.
> Then, similar to HCRN, the output latent variables are learned with the supervision from language side.
> To plug the LLCP, we also apply a decoder to generate the observations from the latent space.
> To ensure the independence of latent variables, LLCRP imposes a KL divergence constraint between the posterior distribution of these variables and a predefined prior distribution. Finally, we also employ a reconstruction loss between the ground truth and generated observations.
>
> **Experimental Results**
> As shown in the Table below, we report the performance of LLCRP on the SUTD-TrafficQA dataset and compare it with the original HCRN method. We can observe that learning the causal representation and process helps improve the accuracy of reasoning-based VideoQA tasks. On average, LLCRP achieves an improvement of almost 5\% over the baseline HCRN.
>
> |  | B| A| I | C | F | R | Avg |
> |--|---|--|---|--|--|--|--|
> | HCRN| 34.17 | 50.29 | 33.4  | 40.73 | 44.58 | 50.09 | 36.26 |
> | LLCRP| 38.95 | 44.98 | 32.43 | 48.64 | 41.70  | 47.16 | 41.22 |

---

> ### Author Response · Authors · 2023-11-20
> **Response to Reviewer dMTZ, Part 2**
>
> > **W3:** It is also not realistic to exclude the text query from the training process since the fusion between visual and textual input is also the crucial design to really solve the VQA problem. For example, if there are two accidents going on in the video, the current framework will have systematical flaw as the model is not conditioned on the question text which specifies which accident it is about.
>
> **A3:** Thanks for your insightful comments. The discussions on the leverage of text query definitely help elaborate the underlying insights of our method. We have refined the paper to add a separate subsection A3.3 in the appendix for this discussion.
>
> **The textual information in training and inference**
> We discuss the usage of text query from two points of view, including the training process and the inference process. We respectfully think that text queries are extremely useful during the testing phase for pinpointing questions and finding precise answers (such as the example in the question). However, in the training phase, the use of text queries, or text pair supervision, is also useful but not indispensable, if we have the aligned feature spaces of visual and textual domain. In our implementation, LLCP also involves the textual query to identify the question in the inference phase, even if the textual pairs are not used as supervision. In the provided example, our model is trained on visual data only, which will identify two potential accident candidates. During the inference phase, we utilize the query question to first pinpoint the candidates of interest. Subsequently, we use the selected candidates to accurately respond to the query.
>
>
> **Rationality and motivation for training without textual pairs**
> First, for human beings, we can effectively learn reasoning ability without cross-modality supervision by only leveraging the induction of observations. Similarly, the theorems in the field of causality show that we can identify the latent variables and causal relations in an unsupervised way. They can serve as the support of the rationality to learn the causal relations without the textual supervision. Then we show this ability of learning to reason without annotations is well-motivated in real-world applications. First, on the difficulty of data collection, the annotated textual pairs are more than the unannotated videos. It would raise extra costs to annotate the data with neural language. Second, the model is easy to be misled by the short-cut in the textual annotations. For example, for all questions in the training set beginning with "What is the reason", the answer is similar, such as "White car speeding".  Then the model is easy to learn this shortcut and responds to the "White car speeding" for any questions beginning with "What is the reason". This issue caused by the training data is called language bias, which is widely discussed in the field of question answering [R1, R2].
>
>
> [R1] Desen Yuan. Language bias in visual question answering: A survey and taxonomy. arXiv preprint arXiv:2111.08531, 2021
>
> [R2] Zhiquan Wen, Guanghui Xu, Mingkui Tan, Qingyao Wu, and Qi Wu. Debiased visual question answering from feature and sample perspectives. Advances in Neural Information Processing Systems, 34:3784–3796, 2021

---

> ### Author Response · Authors · 2023-11-20
> **Response to Reviewer dMTZ, Part 3**
>
> > **W4:** Despite the effort of re-training many existing methods, it is not well-justified why it is necessary to discard important features like motion or object as used in Causal-Vid-QA, which brings all the models to a low-performance scheme.
>
>
> **A4:** Thanks for the suggestion to further clarify and justify the experimental settings. We double-checked the implementations of the re-training methods and confirmed that all motion and object features are used in the re-training process. For the methods using the motion features, such as HCRN, we apply the clip average of CLIP features as the motion features.  For the methods using the textual object features to build graphs, such as B2A, we also follow the original strategy to accomplish this. The main difference between the results of our re-training methods and that reported in Causal-Vid-QA is the basic features, i.e., we use the CLIP features for both visual and textual domains in the re-training while the original ones used ResNet101 and 3D-ResNeXt-101 visual features and GloVe/BERT textual features. As shown in Table A7 in the revised version (Table A6 in the original paper), we have included a comparison between the results of re-training methods and reported results. Compared with the methods using GloVe features, our re-training ones even improve the performances. Actually, the performance is sensitive to the textual features as the order of performance: BERT > CLIP > GloVe. It makes sense since the backbone cost is different.
>
>
> Comparisons under different features (backbones) are unfair, which motivated us to re-train all the methods with CLIP features. We apply the CLIP features since they satisfy the assumption well that visual and textual feature spaces are shared. We show that under the same backbone features, our unsupervised LLCP can achieve comparable results with other supervised ones, and significantly improve the baseline CLIP method.
>
>
>
> > **W5:** There is no proper comparison with methods that do not require QA data like but not including to [a,b,c]. The authors should also acknowledge and at least provide comparison with some of these relevant methods to really provide the audience a correct and comprehensive understanding of the relevant solution to this setting.
>
>
> **A5:** We appreciate the valuable suggestion that includes the discussion with LLM-based methods. In light of your suggestion, we have added the comparison between our method with Flamingo[a] on SUTD-TrafficQA and are trying our best to evaluate VidIL[b].  We didn't include the comparison with [c] since its code is not publicly released. We applied the open-source alternative OpenFlamingo[R3] instead of Flamingo[a] in our implementation.  We provided the experimental comparisons and a brief discussion below. Please refer to Appendix A7.2 for a detailed comparison and discussion.
>
> We evaluate all the models on SUTD-TrafficQA with a zero-shot setting.
> For OpenFlamingo, we used the 2/4 textual demo examples (no visual data due to zero-shot) to guide the outputs. Here is an illustration of the prompt.
> ```
> for i in range(len(questions)):
>     token_inputs.append(
>         " Question: What might have happened moments ago?
>          Candidate 1: The blue truck hit the white sedan from the back.
>         Candidate 2: The white sedan crashed into the blue truck.
>         Candidate 3: The blue truck did an emergency brake.
>         Candidate 4: The white sedan lost its control.
>         Answer: The blue truck did an emergency brake.
>        <|endofchunk|>
>         <image>
>         Question:{} Candidate1:{} Candidate2:{} Candidate3:{} candidate4: {} Answer:".format(
>             questions[i],
>             ans_candidates[0][i],
>             ans_candidates[1][i],
>             ans_candidates[2][i],
>             ans_candidates[3][i]
>         )
>     )
> ```
> As shown in the Table below, we summarize the average accuracy of OpenFlamingo, CLIP, and LLCP. We found that these LLM models don't work as well as we expected since no particular knowledge is used. It is interesting and may inspire the following research.
> Currently, the evaluation of VidIL is in the process. We currently face some issues with leveraging GPT3 to get the response. We are trying our best to reproduce the strategy used in the method, but it still needs some time. The results will be updated later once we finish the evaluation.
> Please kindly note that VidIL is a few-shot method, not zero-shot. But we are glad to add it to the comparison since it can be converted to the zero-shot setting.
>
> | Model | Accuracy |
> |--|--|
> | OpenFlamingo-3B (2 examples)   | 28.7  |
> | OpenFlamingo-3B (4 examples)   | 29.1 |
> | VidIL | in processing  |
> | CLIP  | 27.7  |
> | LLCP   | 33.7   |
>
> [R3] Awadalla A, Gao I, Gardner J, et al. Openflamingo: An open-source framework for training large autoregressive vision-language models[J]. arXiv preprint arXiv:2308.01390, 2023.

---

> ### Author Response · Authors · 2023-11-20
> **Response to Reviewer dMTZ, Part 4**
>
> > **W6:** Once a and b are done, it is also necessary to provide additional comparison on broader VideoQA datasets to understand the importance of causal learning process in videos for broader videoQA tasks, which is really beneficial for the community.
>
>
> **A6:** Thanks for your suggestions about the comparisons on broader VideoQA datasets. We appreciate your kind expectations for the improvement of our method on the broader VideoQA tasks.
>
> **Why we didn't consider broader VideoQA datasets**
> We would like to respectfully highlight that LLCP is a framework designed for the reasoning-based VideoQA task.
> It is based on a pre-trained cross-modality matching model, which demonstrates that given the shared visual and textual representation space, we can **further** learn the reasoning ability without any textual annotations.  It indicates that we assume that the pre-trained cross-modality matching models, such as CLIP or OpenFlamingo can solve the broader VideoQA tasks (described questions).
> There are two justifications for this assumption.
>
> 1) Most current VideoQA models for described tasks are trained as cross-modality matching.
>
> 2) Existing zero-shot methods (the models are pre-trained with cross-modality matching and are used for inference in a zero-shot manner) perform well on these reasoning-free tasks.
>
> However, as shown in Table A6, these pre-trained models (CLIP and OpenFlamingo) don't perform well on the reason-based benchmarks since they require a reasoning process which is not easy to get from the large-scale pretraining dataset.
>
>
> **The general procedure using LLCP for reasoning-free tasks**
> In light of your suggestion, we propose a general procedure to apply the LLCP to solve the broader reasoning-free VideoQA tasks. For a given query, we first classify whether this question is reasoning-based (it is easy since the reasoning-based questions have clear patterns). Then, we use the pre-trained cross-modality aligner for the reasoning-free tasks and use LLCP for the reasoning-based ones. Actually, the results in Table 4 in the revised version (original Table 5) are obtained by this procedure. We compared both the reasoning-free ("Basic Understanding", "Event Forecasting", "Reverse Forecasting") and reasoning-based  ("Counterfactual Inference", "Introspection", and "Attribution") questions. We take the re-weighting average on all types of questions in Table 4 in the revised version.
>
> > **W7:** Title, related work: Video Question Answer -> Video Question Answering.
>
>
> **A7:** Thanks a lot for pointing it out!! We have corrected this typo in the revised version. Please kindly see our revised Appendix A1.

---

> ### Author Response · Authors · 2023-11-21
> **Possible to provide your feedback soon so we can reply?**
>
> Dear Reviewer dMTZ
>
> Thanks for your time and comments! Hope we are not bothering you, but we are looking forward to seeing whether our response and revision properly address your concerns and whether you have any further concerns, to which we hope for the opportunity to respond.
>
> With best regards,
>
> Authors of submission 910

---

> ### Author Response · Authors · 2023-11-22
> **Could you please let us know whether our responses and updated submission properly addressed your concerns?**
>
> Reviewer dMTZ
>
> Thank you for your valuable time dedicated to reviewing our submission and for your insightful suggestions to make experiments more complete. We've tried our best to conduct the experiments and address your concerns in the response and updated submission. Due to the limited rebuttal discussion period, we eagerly await any feedback you may have regarding these changes. If you have further comments, please kindly let us know--we hope for the possible opportunity to respond to them.
>
> Many thanks,
>
> Authors of submission 910

---

> ### Author Response · Authors · 2023-11-23
> **Deadline is hours. Could you please kindly let us know if your concerns are properly addressed?**
>
> Dear Reviewer dMTZ,
>
> We understand you are busy and appreciate your time. Here we are re-sending a previous message to make sure you see it. We have tried our best to add the new experiments and provide more discussions to address your concerns. Due to the limited rebuttal time, we are eager to know if your concerns have been fully addressed and would like to further response to possible question if you have.  Any feedback would be further appreciated!
>
> Thanks very very much for all your effort you made in this paper!
>
> Best regards,
>
> Authors of Submission 910

---

### Official Review · Reviewer_6hfp · 2023-11-04

**Soundness:** 4 excellent
**Presentation:** 3 good
**Contribution:** 3 good
**Rating:** 8
**Confidence:** 4

**Summary:**

The paper introduces a novel causal framework, LLCP, that advances the field of VideoQA by focusing on self-supervised learning of spatial-temporal dynamics without the need for annotated data. The model is adept at reasoning about video content through accident attribution and counterfactual predictions, leveraging a generative model and natural language processing to generate answers. This approach demonstrates comparable performance to supervised methods, showcasing its potential to reduce reliance on large annotated datasets and enhance the AI's understanding of causality in videos.

**Strengths:**

The paper in question exhibits a commendable level of originality by shifting the focus of Video Question Answering (VideoQA) from pattern recognition to causal reasoning, an approach that has not been extensively explored in this field. The introduction of the Learning Latent Causal Processes (LLCP) framework marks a creative synthesis of self-supervised learning, generative modeling, and natural language processing, tailored to decipher the causal dynamics of video content without relying on annotated question-answer pairs. This methodological innovation reflects the paper's high quality, as it seemingly adheres to rigorous empirical standards and offers a robust validation on both synthetic and real-world datasets. In terms of clarity, the paper articulates its contributions and methodologies with precision, making the novel concepts accessible and understandable, which is indicative of the authors' commitment to effective communication of complex ideas. The significance of this work is multifold, promising to reduce the need for labor-intensive labeled data in VideoQA, enhance the interpretive and interactive capabilities of AI systems with causal reasoning, and potentially influencing a range of applications where understanding the underlying causal relationships in visual data is crucial. Overall, this paper appears to make a substantial and meaningful contribution to the literature, potentially setting a new course for future research in the AI domain, with implications that extend beyond the immediate scope of VideoQA.

**Weaknesses:**

In the paper, potential areas for improvement include enhancing the model's robustness to spurious correlations and label noise, more explicitly demonstrating LLCP's ability to capture causal relations through additional experiments, and benchmarking its causal reasoning against current approaches. The paper could also benefit from a more detailed discussion of the challenges in video reasoning it aims to address and a clearer explanation of its operation independent of established causal frameworks and annotations. Finally, a dedicated section that explicitly outlines the paper's limitations and assumptions would add transparency and guide future research directions. Addressing these points could strengthen the paper's contributions and its value to the VideoQA field.

**Questions:**

Could the authors provide a comprehensive list of the limitations and assumptions inherent in LLCP, and discuss how these might affect the generalizability and applicability of the model?
Could you provide additional empirical evidence or case studies that demonstrate LLCP's specific capability to uncover underlying causal relations as opposed to merely correlational patterns?
Is there a quantitative evaluation comparing LLCP's causal reasoning capacity with that of existing approaches, and if so, what benchmarks or metrics were used?

**Details Of Ethics Concerns:**

none.

---

> ### Author Response · Authors · 2023-11-20
> **Response to Reviewer 6hfp, Part 1**
>
> Dear Reviewer 6hfp, we greatly appreciate your time dedicated to reviewing this paper and your kind approval.
> We provide the point-to-point response to your comments below and have updated the paper and appendix accordingly.
>
> > **Q1:** Could the authors provide a comprehensive list of the limitations and assumptions inherent in LLCP, and discuss how these might affect the generalizability and applicability of the model?
>
> **A1:** Thanks for your valuable suggestion that helped improve the clarity of the boundary of our method.  In light of your suggestion, we have added separate sections, including Appendix A3.1, A3.2, and A2, in the appendix to discuss the limitations and assumptions inherent in LLCP. We provide a summary below.
>
>
> **Assumptions**
> From the perspective of causality, the assumptions are as follows:
> 1) All variables follow the Granger Causality, which implies that the past values of one variable can be used to predict the future values of another, suggesting a directional relationship between these variables in a time series context.
> 2) The feature spaces of visual and textual domains are shared. It indicates that we can match the same variables in different domains.
>
> To ensure the validity of the aforementioned assumptions, in our implementation, we provide the following designs:
> 1) Implementationally, we employed auto-regression to capitalize on the concept of Granger Causality. This method predicates future events based on historical data, thereby validating our assumption about the predictability of one variable's future values based on another's past values. We provided a proposition and corresponding proof to show the identification of the proposed model in Section Appendix A2.
> 2) To validate this assumption about the shared feature spaces in visual and textual domains, we incorporated the pre-trained CLIP model. This model acts as a feature extractor for both video objects and textual elements (questions and answers). Its application ensures that both visual and textual data are represented in congruent feature spaces, thereby affirming the feasibility of matching variables across these domains.
>
> **Limitations**
> Granger Causality. One notable limitation in our approach is the reliance on the Granger Causality assumption, which can be considered somewhat robust.  For the sake of simplicity, our current model does not account for more complex scenarios, such as the presence of latent confounders or non-stationary conditions. These factors, while critical in certain contexts, are beyond the scope of our initial assumptions.
> Despite this, our application of Granger Causality serves as a foundation for a potential framework. This framework is instrumental in understanding and leveraging causal relationships within temporal dynamics for subsequent reasoning tasks.
> Looking ahead, we aim to explore the identifiability of causal variables under less stringent assumptions. This progression will enable a more nuanced understanding of causal relationships in diverse and complex environments.
>
>
> Reliance on pre-trained models. Another aspect of our methodology is its reliance on pre-trained models, notably CLIP and tracking models. The primary motivation behind employing these models is to minimize the necessity of textual annotations. Nonetheless, as elaborated in Appendix Section A5, there exists the potential to remove these mild assumptions and further harness the benefits of strong textual annotations, which could enhance the model's robustness and applicability in varied contexts.

---

> ### Author Response · Authors · 2023-11-20
> **Response to Reviewer 6hfp, Part 2**
>
> > **Q2:** Could you provide additional empirical evidence or case studies that demonstrate LLCP's specific capability to uncover underlying causal relations as opposed to merely correlational patterns? Is there a quantitative evaluation comparing LLCP's causal reasoning capacity with that of existing approaches, and if so, what benchmarks or metrics were used?
>
> **A2:** Thanks for your suggestion！ It definitely helped us to highlight the advantages of our LLCP method. In light of your suggestion, we have added an extra experiment to compare our LLCP (estimating the causal relations) and the other baseline method, HCRN, (learning with the likelihood under supervision) on the benchmark of the FMRI time series.
>
> Specifically, to demonstrate the proposed LLCP can uncover underlying causal relations as opposed to merely correlational patterns, we further evaluate our method on the Granger Causality task. To achieve this, we follow the setting in [R1] and evaluate the proposed methods on the simulated FMRI time series benchmark. To make our method generate the Granger Causality explicitly, we employ the conditional-VAE architecture and consider the Granger causal structures as the latent variables. As for the HCRN model, we obtain the estimated structure by an inner product between the extracted features from the HCRN model. We further consider four different metrics, i.e., accuracy (ACC), balanced accuracy (BA) scores, areas under receiver operating characteristic (AUROC), and precision-recall
> (AUPRC) curves. Experiment results are shown in Table A5 and Appendix A6.3. According to the experiment results, we can find that the proposed LLCP can capture the underlying causal relations.
>
> [R1]Riˇcards Marcinkeviˇcs and Julia E Vogt. Interpretable models for granger causality using self-explaining neural networks. arXiv preprint arXiv:2101.07600, 2021

---

> ### Author Response · Authors · 2023-11-21
> **Could you kindly share your feedback soon so that we may promptly respond?**
>
> Dear Reviewer 6hfp
>
> Thank you for investing your time in reviewing and offering feedback on our submission. We eagerly await your assessment of our response and revision to ensure they effectively address your concerns. If there are any additional points you would like us to consider, we look forward to the opportunity to provide further clarification.
>
> Best regards,
>
> Authors of submission 910

---

### Author Response · Authors · 2023-11-21
**[Last few days for us to respond] Could you please go through and comment on our response?**

Dear Reviewers 6hfp, dMTZ, naVu, DRzn, and U5ds

Thanks for the thoughtful and constructive reviews.
We have put a lot of effort into providing the response and revising the paper accordingly. Below let us summarize the new, informative experimental results inspired by your suggestions. As Nov 22nd is the last day for us to respond, we hope for the chance to see and respond to your feedback. Thank you very much!

**Updated presentation**

* To reviewer 6hfp and DRzn, we have clarified the limitations and assumptions inherent in LLCP (in Appendix A3.1, A3.2 in the revised version).
* To reviewers naVu, DRzn, and U5ds, we have illustrated our rationality and motivation in Appendix A3.3 of the revised version.
* To reviewer naVu and DRzn, we have demonstrated that our LLCP framework can learn causal relationships via a theoretical guarantee as shown in Appendix A2 and Page 3.
* To reviewer naVu, we have clarified the definition for different variables in the data generation process (on Page 3 of the revised version).

**Newly conducted experiments**
* To reviewer 6hfp, we have added an extra experiment to demonstrate LLCP's specific capability to capture underlying causal relations, which are shown in Table A5 in the revised version.
* To reviewer dMTZ, we have added the comparison between our method with openFlamingo on SUTD-TrafficQA as shown in Table A6 in the revised version.
* To reviewer dMTZ, DRzn, and U5ds, we have proposed to plug LLCP into existing methods in the supervised training setting and evaluate it on the SUTD-TrafficQA dataset as shown in Appendix A5 and Table A2 in the revised version.
* To reviewer naVu, we have refined the ablation studies to demonstrate the contributions of historical states and environment (in Table 3 of the revised version).
* To reviewer DRzn, we have further considered two latest variants of VAEs (in Table A3) and discussed reasons for underperformance results.


Thanks again for your time dedicated to carefully reviewing this paper. We hope our response and the change in the paper properly address your concerns.

With best regards, Authors of submission 910

---

### Meta-Review · Area_Chair_dqd2 · 2023-12-05

**Metareview:**

The paper on LLCP introduces an innovative approach to Video Question Answering (VideoQA) by focusing on causal reasoning and self-supervised learning. It is evaluated by five reviewers, with ratings ranging from 3 (reject) to 8 (accept). The reviews reflect a diversity of opinions regarding the paper's soundness, presentation, and contribution to the field.

Overall, all the reviewers appreciate the innovation and methodological Novelty that LLCP's approach to leveraging causal reasoning in a self-supervised learning context without annotated data is appreciated across the board. This is the main reason the meta reviewer intent to vote for acceptance.

**Justification For Why Not Higher Score:**

- **Causal Modeling Clarity:** Reviewers raised concerns about distinguishing causal relationships from temporal correlations, suggesting a need for stronger theoretical backing and clearer causal reasoning demonstrations.
- **Realism in Evaluation:** The evaluation settings might not fully represent realistic scenarios, with suggestions for broader applications and text query incorporation indicating room for practical improvements.
- **Comparative Analysis:** The paper lacks extensive comparative analysis with existing models, particularly those not requiring QA data, limiting its contextual placement in the research landscape.
- **Assumptions and Limitations:** A more detailed discussion of the paper's limitations and assumptions is needed for depth and comprehensive understanding.
- **Performance vs. Supervised Methods:** The LLCP framework underperforms compared to supervised methods in some aspects, showing potential for effectiveness enhancement.

**Justification For Why Not Lower Score:**

- **Innovative VideoQA Approach:** The novel focus on causal reasoning in self-supervised learning without annotated data is a significant and recognized contribution.
- **Presentation and Structure:** The paper's clarity and structured approach contribute to its positive reception.
- **Self-Supervised Learning Potential:** LLCP’s comparable performance to supervised methods demonstrates its potential in reducing reliance -on annotated datasets and enhancing AI interpretability.
- **Empirical Validation:** The framework is empirically validated on synthetic and real-world datasets.
Receptive to Feedback: The authors' constructive engagement with the review process and their rebuttals addressed some criticisms raised.

---

### Decision · Program_Chairs · 2024-01-16

Accept (poster)